

# Shifts of prepotentials

**Nikita Nekrasov[1,2,3], Nicolo Piazzalunga[4] and Maxim Zabzine[4],**
with an appendix by **Michèle Vergne[5]**

**1** Simons Center for Geometry and Physics, SUNY Stony Brook, NY 11794-3636, USA
**2** Center for Advanced Studies,[1] Skoltech, Moscow, Russia
**3** Kharkevich Institute for Information Transmission Problems,[2] Moscow, Russia
**4** Department of Physics and Astronomy, Uppsala University,
Box 516, SE-75120 Uppsala, Sweden
**5** Université Paris 7 Diderot, Institut Mathématique de Jussieu,
Sophie Germain, case 75205, Paris Cedex 13

## Abstract

We study the dynamics of supersymmetric theories in five dimensions obtained by compactifications of $M$-theory on a Calabi-Yau threefold $X$. For a compact $X$, this is determined by the geometry of $X$, in particular the Kähler class dependence of the volume of $X$ determines the effective couplings of vector multiplets. Rigid supersymmetry emerges in the limit of divergent volume, prompting the study of the structure of Duistermaat-Heckman formula and its generalizations for non-compact toric Kähler manifolds. Our main tool is the set of finite-difference equations obeyed by equivariant volumes and their quantum versions. We also discuss a physical application of these equations in the context of seven-dimensional gauge theories, extending and clarifying our previous results. The appendix by M. Vergne provides an alternative local proof of the shift equation.

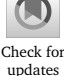

## Contents

---

[1]visiting professor
[2]on leave of absence

# 1 Introduction and summary of results

In realizations of topological, or partially topological, field theories via supersymmetric field theories, one often encounters the paradigm of "fields, equations, symmetries" as the basic setup [1] of an enumerative moduli problem, for which the supersymmetric field theory provides an integral representation. The linearized equations and the linearized symmetry transformations define the differentials in a three-term complex $\mathcal{T}$, whose cohomology $H^0(\mathcal{T})$ in degree zero describes infinitesimal automorphisms, the cohomology in degree one $H^1(\mathcal{T})$ describes infinitesimal deformations, while $H^2(\mathcal{T})$ packs the obstructions to deformations (the first order deformations may not extend to the second order, the obstruction is given by a quadratic Kuranishi map from $H^1(\mathcal{T})$ to $H^2(\mathcal{T})$).

Physically, the zero modes of fermions in the supermultiplet of a cohomological field theory belong to these cohomology groups. The contribution to the path integral of a specific point in the moduli space is non-zero if the Yukawa interactions of the fermions and bosons saturate the fermionic zero modes, while the integral over the bosonic modes is finite. The latter is problematic if the moduli space in question is non-compact. The non-compactness of the moduli space of solutions to the partial differential equations representing, e.g., the BPS equations could be both of ultra-violet (such as point-like instantons) and of infra-red (such as the runaway of a localized solution to infinity in space-time) nature.

While ultraviolet noncompactness requires knowledge of microscopic degrees of freedom of the theory, in practice one fixes it by modifying the theory at short distances in a controllable way. For example, one uses noncommutativity in gauge theories with unitary gauge groups, or coupling to gravity in two-dimensional sigma models.

Infrared noncompactness is partly cured by $\Omega$-deformation [2]. Mathematically, this means working equivariantly with respect to some rotational symmetries, or, more generally, isometries of the background. Localization techniques such as Duistermaat-Heckman formula and Berline-Vergne-Atiyah-Bott theorem are ubiquitous in mathematics and physics, and play key roles in fields as diverse as enumerative geometry [3–5], exact calculations of effective actions and BPS-protected correlators [2,6,7], gauge/string [8] and quantum gravity/topological string [9] dualities, calculations of black hole entropy [10], and $a$-maximization [11,12].

Despite the wide applications, the justification of the equivariant approach and the precision of its predictions for a given quantum field theory is not always clear. There are numerous puzzles related to the importance of boundary terms [13,14], decoupling or non-decoupling of gravity modes, (non)existence of global symmetries [15], for which one turns on chemical potentials in the twisted Witten index [16].

In this paper we shall be discussing $M$-theory compactifications on a Calabi-Yau threefold $X$ times five-dimensional Minkowski space $\mathbb{R}^{1,4}$, or further compactifications of the form $\mathcal{Y} \times \mathbb{R}^1$, with a Calabi-Yau five-fold $\mathcal{Y}$. Moreover, we shall also study the limits where the manifolds $X$ or $\mathcal{Y}$ have infinite volumes, effectively decoupling (super)gravity in five or one dimension. More precisely, by working in a limit of low energy in five (one) dimensions, one does not excite eleven-dimensional (super)gravity modes, which effectively are frozen in the background. But let us start with the case of compact $X$.

The classical $\mathcal{N} = 1$ five-dimensional supergravity theory, obtained by compactification of eleven-dimensional supergravity on a Calabi-Yau threefold $X$, contains $h^{1,1}(X) - 1$ vector multiplets, whose low energy dynamics is governed by the superspace action, derived from the homogeneous degree-three prepotential [17,18]

$$\mathcal{F}(T) = \frac{1}{3!} \sum_{a,b,c=1}^{h^{1,1}(X)} c_{abc} T^a T^b T^c \,, \tag{1.1}$$

where $T^a$ are linear coordinates on $H^{1,1}(X)$ that we specify below. The matrix $c_{abc}$ is the matrix of the triple intersection form, so that in fact eq. (1.1) is the symplectic volume of $X$

$$\mathcal{F}(T) = \int_X e^k \,. \tag{1.2}$$

Here $k$ is the Kähler form associated to the Calabi-Yau metric on $X$, which can be expanded in some integral basis $(\omega_a)$ of $H^2(X)$ (recall that for proper Calabi-Yaus $h^{2,0} = h^{0,2} = 0$)

$$k = \sum_{a=1}^{h^{1,1}(X)} T^a \omega_a \,, \qquad \omega_a \in H^2(X, \mathbb{Z}) \cap H^{1,1}(X) \,. \tag{1.3}$$

As it stands, eq. (1.2) is a formal expression where one understands that the exponential of a two-form is expanded in Taylor series, and only the six-form can be integrated over $X$, the rest of the series integrating to zero. However, in extending the de Rham differential to the so-called equivariant derivative, one appreciates the use of the exponential form of the integral eq. (1.2) in that it leads to a streamlined formulation of certain *WKB-exact* integration formulas.

In writing eq. (1.2) or eq. (1.1) one uses a redundant set of parameters, the homogeneous coordinates on the Coulomb branch of the moduli space of vacua, which is parametrized by the vacuum expectation values of the real scalars in the vector multiplets, which, as we said

earlier, come in the amount of $n_V = h^{1,1}(X) - 1$. The number of $U(1)$ gauge fields, however, is $h^{1,1}(X)$, as these are obtained by Kaluza-Klein (KK) decomposing the three-form $C^{(3)}$ of eleven-dimensional supergravity

$$C^{(3)} = \sum_{a=1}^{h^{1,1}(X)} A^a \wedge \omega_a \,, \tag{1.4}$$

where in the KK approximation the $\omega_a$ are integral harmonic two-forms on $X$, while $A^a$ are the $U(1)$ gauge fields propagating in $\mathbb{R}^{1,4}$. The subtlety of five-dimensional supergravity is that a specific linear combination of $U(1)$ vector fields in eq. (1.4), the graviphoton, is in the supermultiplet of graviton. The remaining $U(1)$ fields fall in the supermultiplets containing the real scalars in the moduli space $\mathcal{M}_V$, which is locally parametrized by the scalars $T^a$, restricted by the relation

$$\mathcal{F}(T) = -i\Omega \wedge \bar{\Omega} \tag{1.5}$$

set up by another part of the Calabi-Yau data, the normalization of the holomorphic $(3,0)$-form $\Omega$. Together with the choice of the harmonic 3-form (flat 3-form field) on $X$ and their superpartners, they form the massless hypermultiplet content of the effective five-dimensional theory. The tangent space to the hypermultiplet factor $\mathcal{M}_H$ in the scalar manifold of the five-dimensional theory coincides with the space of infinitesimal deformations of the Calabi-Yau structure, preserving the Kähler structure, compatible with $\Omega$.

In making $X$ non-compact we face the difficult problem of finding the $L^2$-spectrum of the Hodge Laplacian acting on the space of two-forms. Such harmonic forms enter eq. (1.4) to define the dynamical gauge fields propagating in five dimensions. The total number of linearly independent harmonic two-forms, not necessarily $L^2$-normalizable, is in general larger. We can interpret the excess as corresponding to non-dynamical vector fields, which can be viewed as flavor symmetries of the effective five-dimensional supersymmetric theory. For example, in $SU(2)$ gauge theory in five dimensions, the vacuum expectation value $\varphi\sigma_3 = \langle\phi\rangle$ of the real scalar $\phi$ in the vector multiplet breaks the gauge symmetry to its maximal torus $U(1)$. The $SU(2)$ instantons, which are solitonic particles, have a mass given by the sum of the inverse renormalized coupling squared $g^{-2}$ and the absolute value $|\varphi|$. This agrees with a BPS formula where $g^{-2}$ is interpreted as a scalar in a vector multiplet of some flavor *topological* symmetry (the gauge field $\theta$ in that multiplet would give rise to the coupling $\theta \wedge \operatorname{tr} F \wedge F$ producing the theta term in four dimensions, if the theory is compactified further on a circle). The theory also has a BPS particle, the $W$-boson, of mass $|\varphi|$.

The gauge symmetry and the *topological* symmetry correspond, in a realization of $SU(2)$ theory as $M$-theory "compactified" on the total space $X$ of the $\mathcal{O}(-2,-2)$ line bundle over $\mathbb{P}^1 \times \mathbb{P}^1$, to the degree-two cohomology of $X$. As the latter can be contracted to the product of two spheres, the degree-two cohomology is two-dimensional, it is generated by the first Chern classes $x_1, x_2$ of the line bundles $\mathcal{O}(1)$ over the first and second $\mathbb{P}^1$ factors, respectively. The manifold $X$ has, additionally, one-dimensional degree-four cohomology, generated by $x_1 x_2$. Its Poincare-dual can be represented by a two-form with compact support. The latter acts on $H^2(X)$ by shifts. In the $M$-theory realization the $W$-boson and instanton particles are $M2$-branes wrapping the two $\mathbb{P}^1$ factors in $\mathbb{P}^1 \times \mathbb{P}^1$. Which one is the $W$-boson and which one is an instanton depends on the relative size of the two cycles. The presence of the $|\varphi|$ term in the formulas for both masses reflects the action of $H^2_{\text{comp}}(X)$ on $H^2(X)$.

Now we come to the main point of this paper. In most geometric realizations of quantum field theories in five dimensions (and their compactified four-dimensional versions) the "internal" Calabi-Yau manifold $X$ is not only non-compact, it is a toric manifold. For toric manifolds $X$, there are additional non-dynamical symmetries in the effective five-dimensional theory. Indeed, any isometry of $X$ leads to a Kaluza-Klein-like gauge field in five dimensions. Thus, $H^1(\mathcal{T})$

of the deformation complex corresponds to the tangent space to the space of scalars, while $H^0(\mathcal{T})$ corresponds to gauge fields in the effective five-dimensional theory, both dynamical and non-dynamical. We leave to further study the interpretation of $H^2(\mathcal{T})$ cohomology (for compact $X$ the theorem of Tian-Todorov says that the obstructions vanish, $H^2(\mathcal{T}) = 0$). Presently, let us focus on the non-generic enhancement of $H^0(\mathcal{T})$ thanks to the isometries of $X$.

Consider the gravitational background of the form $X \times Y$, with internal manifold $X$ with metric $g = g_{mn}(x)dx^m dx^n$ and spacetime $Y$ with metric $h = h_{\mu\nu}(y)dy^\mu dy^\nu$. Suppose the Lie group $G$ acts on $(X, g)$ by isometries, generated by vector fields $V_a = V_a^m(x)\frac{\partial}{\partial x^m}$, for $a = 1, \dots, \dim G$. The Kaluza-Klein ansatz gives us gauge fields $A^a = A_\mu^a(y)dy^\mu$ on $Y$. Geometrically, turning on these gauge fields means modifying $X \times Y$ to the total space $Z$ of a locally trivial bundle $Z \to Y$, with the $G$ connection $A$ that features in the metric on $Z$:

$$ds^2 = h_{\mu\nu}(y)dy^\mu dy^\nu + g_{mn}(x)\Big(dx^m + V_a^m(x)A_\mu^a(y)dy^\mu\Big)\Big(dx^n + V_b^n(x)A_\nu^b(y)dy^\nu\Big). \quad (1.6)$$

For compact $X$ the gauge fields $A^a$ are dynamical, and the Einstein-Hilbert action gives, in the limit of small volume of $X$, the Yang-Mills action with gauge group $G$. Even if the manifold $X$ is non-compact, so that the gauge fields $A^a$ are non-dynamical, they can be turned on as background fields.

In our present setup, of all the isometries of $X$ we need those which preserve, in addition to the metric $g$, the covariantly constant spinors, generating rigid supersymmetry in five spacetime dimensions. Practically, for toric $X$ we are taking the generators of the torus, preserving the holomorphic top-degree form. There are two linearly independent vector fields for Calabi-Yau threefolds, let us call them $V_1$ and $V_2$. The associated five-dimensional background gauge fields $A^1$ and $A^2$ belong to the vector multiplets. Their lowest components are the real scalars $\varepsilon^1, \varepsilon^2$. Without breaking five-dimensional super-Poincare invariance, we can deform the theory by turning on a non-zero value of these scalars. The so-deformed theory deserves a separate study. Presently we assume this deformation corresponds to the equivariant extension of differential forms that we employ below.

Once the five-dimensional theory is compactified on a circle, in addition to the real scalars $\varepsilon^{1,2}$ one can turn on the components of the gauge fields $A^1, A^2$ along the circle. Geometrically it means fibering the Calabi-Yau threefold over the circle, with the isometric twist of the fiber. In other words we take a quotient of $\mathbb{R}^1 \times X$ by the action of $\mathbb{Z}$ by simultaneous translation along $\mathbb{R}$ by $2\pi\beta$ and the isometry transformation of $X$ by $e^{\beta(\varphi^1 V_1 + \varphi^2 V_2)}$. In this way the real parameters $\varepsilon^{1,2}$ get complexified to $q_1, q_2 \in \mathbb{C}^\times$,

$$q_1 = e^{-\varepsilon^1 + i\beta\varphi^1}, \quad q_2 = e^{-\varepsilon^2 + i\beta\varphi^2}. \quad (1.7)$$

One can make an additional twist by the isometry of $X$ not preserving the holomorphic top form, provided we turn on the compensating twist of the remaining space $\mathbb{R}^4$. More generally, we can replace $X \times \mathbb{R}^4$ by some toric Calabi-Yau fivefold $\mathcal{Y}$.

In the earlier work [19], a twisted version of a nonabelian gauge theory in seven dimensions was studied. Mathematically, the theory is the nonabelian rank $n$ K-theoretic Donaldson-Thomas theory on a three-fold (which in ref. [19] was taken to be a toric Calabi-Yau manifold). We are interested in factorization properties of the supersymmetry-protected sector of that theory (fundamentally the theory needs an ultraviolet completion, and the factorization might be sensitive to those details). Since our main tool is supersymmetric localization, in fact, equivariant localization with respect to the isometries of the Calabi-Yau space, we are forced to work on noncompact spaces. This leads to several puzzles.

The physical problem that involves passing to noncompact Calabi-Yau manifolds is the realization of rigid supersymmetric theory as a limit of supergravity, in which the supergravity multiplet freezes, while five-dimensional vector multiplets remain dynamical [20, 21]. This is

usually realized by a specific large $T$ limit of eq. (1.1), where some of the $T^a$ variables go to infinity (essentially, as $M_{\text{Planck}} \to \infty$) while others remain finite. We denote the finite Kähler parameters by $t^a$ in what follows. This leads to the expression

$$\mathcal{F}(T) \to F(t; \Lambda) = \Lambda^2 F_1(t) + \Lambda F_2(t) + F_3(t), \tag{1.8}$$

where the coefficients $F_1, F_2$ of the divergent terms as $\Lambda \to \infty$ are linear and quadratic in $t$. In the field theory interpretation of eq. (1.8) the leading divergence $\propto F_1$ is irrelevant, while the term quadratic in $t$ gives the bare gauge couplings. The cubic term describes the effect of (one-loop) corrections to the effective gauge couplings and five-dimensional Chern-Simons term, induced by the loops of some charged heavy fields.

The 11d origin of eq. (1.1) is the Chern-Simons term in the action of eleven-dimensional supergravity [22]. As is often the case with Chern-Simons terms, it can be expressed as a twelve-dimensional integral [23]

$$\int_{\mathcal{Z}^{12}} G^{(4)} \wedge G^{(4)} \wedge G^{(4)}, \tag{1.9}$$

where $\partial \mathcal{Z}^{12}$ is the eleven-dimensional spacetime. The main case of interest for us is the eleven-dimensional manifold, which is a fibration over a circle $S^1$ with fiber a Calabi-Yau fivefold $\mathcal{Y}$. For toric $\mathcal{Y}$ the twelve-dimensional extension $\mathcal{Z}^{12}$ can be constructed as a Kähler manifold with boundary, which is a holomorphic fibration over a unit disk $|w| \le 1$.

Our goal is to construct equivariant extensions of eq. (1.2) and its higher analogue

$$\mathcal{F}(t, \dots) = \int_{\mathcal{Z}^{12}} \exp G^{(4)} \tag{1.10}$$

and understand the structure of the full answer for non-compact spaces, without making ad hoc choices. Details are provided below.

Now let us clarify the main subtlety. The definition of the triple intersection form in eqs. (1.1) and (1.2) does not make sense for toric Calabi-Yau threefolds, which are always non-compact. Nevertheless, there is abundant literature discussing these intersections in the context of topological strings and 5d gauge theories. The usual logic is the following: geometric intersection numbers when one of the divisors is compact are fixed unambiguously, the rest (when all the divisors involved are non-compact) is fixed in some ad hoc fashion.

As an example, let us look at local $\mathbb{P}^1 \times \mathbb{P}^1$ i.e. the total space of the line bundle $\mathcal{O}(-2,-2) \to \mathbb{P}^1 \times \mathbb{P}^1$. There are two versions of $\mathcal{F}(t)$: an asymmetric version in the context of 5d gauge theories [24, section B.3.1], and a symmetric one in the context of topological strings [25, section 2.1]. These two versions correspond to different ways of fixing the intersection numbers of non-compact divisors. On the other hand, when we try to compute them from first principles via equivariant localization, and the Duistermaat-Heckman (DH) formula, we are unable to reproduce the known results in the literature. This question is not really Calabi-Yau three-fold-specific. Indeed, it applies to all non-compact toric Kähler quotients. The situation is even more confusing since there are examples where, upon certain choices, one reproduces the known intersection matrix as the first regular term in DH expansion (see the example of $A_{n-1}$ spaces [26]) and there are cases where there is no way to get rid of equivariant parameters and extract a polynomial in Kähler parameters.

In this work we study systematically the equivariant DH formula on non-compact toric Kähler manifolds and explain how to deduce geometric information from it. We use equivariant volumes as regularized expressions for $\mathcal{F}(T)$, so that the role of the cutoff parameter $\Lambda$ will be played by the inverse powers $\varepsilon^{-1}$ of the $\Omega$-deformation parameters. This is similar in spirit to the approach of ref. [26]. The crucial new ingredient is the set of difference equations

(shift equations), representing the action of cohomology with compact support on de Rham cohomology, which we use to extract the finite pieces $F_3(t)$ and constrain $F_2(t)$.

In a way, this is similar to the time-tested dispersion relations in quantum field theory, which are used to constrain the loop contributions to Green's functions. The ambiguities in reconstructing the real part of the Green's function given its imaginary part, which is computed by lower-order diagrams, correspond to the ultraviolet counterterms. In our story these ambiguities correspond to the effects of frozen supergravity fields.

We concentrate on manifolds $X$ with non-zero second cohomology with compact support $H^2_{\text{comp}}(X)$, as these are relevant to geometric engineering and our equations are simpler here. Upon certain (non-canonical) choices, we can solve this shift equation and produce the analog of intersection polynomials for non-compact spaces. The non-uniqueness of this polynomial mirrors the relation between $H^2_{\text{comp}}(X)$ and $H^2(X)$. The shift equation helps us structure the full equivariant result in a geometric fashion, without throwing away singular terms. Its physical meaning is to compare two spaces with the same asymptotics but different values of Kähler parameters, in the same chamber, to extract finite data. We also analyze the quantum mechanical analog of our shift equation and the related semi-classical expansion. In this context, the difference of partition functions between singular and resolved spaces is effectively reduced to a quantum mechanical problem on a compact space. We observe that higher-dimensional compact support cohomologies lead to higher-order difference equations for equivariant volumes and partition functions. We apply this set of ideas to toric Calabi-Yau fivefolds, to get the correct classical action for DT theory, in the presence of four-cycles.

Several mathematical works addressed the issue of localization on non-compact spaces, see e.g. the thesis [27] and references therein, as well as refs. [28, 29]. Rather than focusing on technical details, here we try to isolate a simple phenomenon, which surprisingly was overlooked in prior literature, by looking at some key examples. This work is thus in the field of experimental mathematics. Most likely, the action of compact support cohomology on de Rham cohomology can be extended to the equivariant setting, and one should study the action of compact support equivariant cohomology on equivariant cohomology [30, 31]. We believe our constructions can be extended to more complex setups, perhaps even to the full quantum cohomology ring, although the details remain to be worked out, and similarly for the relation of our shift equations to the chamber structure and wall-crossing.

It may be that our ideas play some role in other generalizations, for example we find some resemblance between our eq. (6.6) for the conifold and a similar shift equation for resolved conifold in the context of Gromov-Witten theory [32, 33]. In this context it is crucial to push the Calabi-Yau fivefold picture along the lines we have outlined here, and try to come up with the appropriate quantum mechanical generalization of higher times.

**Plan of the paper** In section 2 we set up the geometric framework for toric Kähler quotients and discuss applications of DH formula. We stress the difference between compact and non-compact cases. In section 3 we provide the quantum mechanical description of toric Kähler quotients and define the equivariant partition function, generalizing DH formula. We discuss the corresponding semi-classical expansion of the partition function, stressing the difference between compact and non-compact settings. Section 4 presents the shift equation, which is a difference equation for the equivariant volume associated to the action of cohomology with compact support on de Rham cohomology. We mostly focus on the case of $H^2$ cohomology. In section 5 we go through different examples in real dimension 4, 6 and 10. In section 6 we briefly discuss the cases of cohomology with compact support in higher degree, in two explicit examples. In section 7 we introduce the notion of higher times for the equivariant volume. We explain how the higher time formalism can be used to write the shift equations corresponding to higher-degree cohomology with compact support. In section 8 we illustrate

the use of higher times for a specific Calabi-Yau fivefold, and apply our techniques to a physically relevant class of problems, namely the factorization of classical actions in the context of non-abelian Donaldson-Thomas theory on Calabi-Yau threefolds. This clarifies and extends the considerations of ref. [19].

An appendix by M. Vergne provides an alternative local proof of the main shift equation.

## 2  Geometric setup

We start with the complex vector space $\mathbb{C}^N$ with complex coordinates $\boldsymbol{z} = \left(z^i\right)_{i=1}^N$ and constant Kähler form in the coordinates $z, \bar{z}$:

$$\omega_0 = \frac{1}{2\mathrm{i}} \sum_{i=1}^N dz^i \wedge d\bar{z}^i . \tag{2.1}$$

Let $\mathbb{T} = U(1)^r$ denote an $r$-dimensional compact torus, and $\mathbb{T}_\mathbb{C} = \left(\mathbb{C}^\times\right)^r$ its complexification.

A unitary $\mathbb{T}$-action on $\mathbb{C}^N$ is determined by an integral map $Q : \mathbb{Z}^r \to \mathbb{Z}^N$, with integer-valued matrix $(Q_i^a)$ for $a = 1, \ldots, r$, and $i = 1, \ldots, N$, the charge matrix in the language of gauged linear sigma models:

$$\boldsymbol{z} \mapsto \left(e^{\mathrm{i}\vartheta_1}, \ldots, e^{\mathrm{i}\vartheta_r}\right) \cdot \boldsymbol{z} = \left(e^{\mathrm{i}Q_i^a \vartheta_a} z^i\right)_{i=1}^N , \tag{2.2}$$

where we sum over repeated indices $a$. The same $Q$ defines the holomorphic action of $\mathbb{T}_\mathbb{C}$ on $\mathbb{C}^N$ – simply let $\vartheta_k$'s in eq. (2.2) be complex. We require that

$$\gcd_i Q_i^a = 1 \quad \text{for every } a . \tag{2.3}$$

Denote by $\boldsymbol{p} = \left(p^i = |z^i|^2\right)_{i=1}^N$ the map $\mathbb{C}^N \to \mathbb{R}_+^N$. The $\mathbb{T}$-action eq. (2.2) defines the moment map $\mu : \mathbb{C}^N \to \mathbb{R}^r = (\mathrm{Lie}\,\mathbb{T})^*$ by

$$\mu = Q^{\mathrm{tr}} \cdot \boldsymbol{p} = (\mu^a)_{a=1}^r , \qquad \mu^a(\boldsymbol{z}, \bar{\boldsymbol{z}}) = \sum_{i=1}^N Q_i^a p^i . \tag{2.4}$$

Fix an $r$-tuple $\boldsymbol{t} = (t^a)_{a=1}^r$ of real numbers $t^a$, then the symplectic quotient is defined as

$$X_{\boldsymbol{t}} = \mu^{-1}(\boldsymbol{t})/\mathbb{T} . \tag{2.5}$$

For generic $\boldsymbol{t}$ with eq. (2.3) in place, $X_{\boldsymbol{t}}$ is a smooth symplectic manifold of dimension $2(N-r)$, with the symplectic form $\varpi_{\boldsymbol{t}}$ verifying:

$$p^* \varpi_{\boldsymbol{t}} = i^* \omega_0 , \tag{2.6}$$

where $p : \mu^{-1}(\boldsymbol{t}) \to \mu^{-1}(\boldsymbol{t})/\mathbb{T}$ is a projection, and $i : \mu^{-1}(\boldsymbol{t}) \longrightarrow \mathbb{C}^N$ the embedding. Moreover $X_{\boldsymbol{t}}$ inherits a complex structure from the identification

$$X_{\boldsymbol{t}} = \left(\mathbb{C}^N\right)^{\mathrm{stable}} / \left(\mathbb{C}^\times\right)^r , \tag{2.7}$$

where the notion of stability is discussed below. Together, the complex and symplectic structures make $X_{\boldsymbol{t}}$ a Kähler manifold of complex dimension $d = N - r$, and $r$-dimensional Picard variety, so that $\dim H_2(X_{\boldsymbol{t}}) = r$. For two closeby values $\boldsymbol{t} \approx \boldsymbol{t}'$, the corresponding manifolds $X_{\boldsymbol{t}}$ and $X_{\boldsymbol{t}'}$ are diffeomorphic, so we can skip the subscript $\boldsymbol{t}$, if we are only interested in the manifold structure.

The symplectic forms $\varpi_t$ on $X_t$ and $\varpi_{t'}$ on $X_{t'}$, upon identification of the manifolds, can be compared. The theorem of Duistermaat and Heckman states that the cohomology class $[\varpi_t]$ depends on $t$ linearly, as long as one does not cross critical values of $\mu$.

An important combination of the charges

$$Q^a := \sum_{i=1}^{N} Q_i^a \tag{2.8}$$

measure the degree of non-invariance of the holomorphic top-degree form on $\mathbb{C}^N$. If

$$Q^a = 0 \tag{2.9}$$

for all $a$, then $X_t$ is also a Calabi-Yau (CY) $d$-fold, necessarily non-compact. Specifically, it means two things:

1. $X_t$ inherits a holomorphic $(d,0)$-form from the top-degree holomorphic form on $\mathbb{C}^N$:

   $$\Omega = \iota_{\mathcal{V}^1} \cdots \iota_{\mathcal{V}^r} dz^1 \wedge \cdots \wedge dz^N , \tag{2.10}$$

   where

   $$\mathcal{V}^a = \sum_{i=1}^{N} Q_i^a z^i \frac{\partial}{\partial z^i} \tag{2.11}$$

   are the holomorphic vector fields on $\mathbb{C}^N$, generating the $\mathbb{T}_{\mathbb{C}}$-action.

2. there exists a smooth function $f : X_t \to \mathbb{R}$, such that the modified Kähler form:

   $$\omega = \varpi_t + \partial \bar{\partial} f \tag{2.12}$$

   defines a Ricci-flat metric:

   $$\omega \wedge \ldots \wedge \omega \propto \Omega \wedge \bar{\Omega}, \tag{2.13}$$

   with some constant prefactor (which may depend on complex moduli).

Although the Calabi-Yau condition is important for supersymmetry considerations, most of our discussion below goes through without it.

Choose a one-parameter subgroup of $\left(\mathbb{C}^\times\right)^N / \mathbb{T}_{\mathbb{C}}$, acting nontrivially on $X_t$. It defines a Hamiltonian $H_\epsilon : X_t \to \mathbb{C}$

$$H_\epsilon = \sum_{i=1}^{N} \epsilon_i p^i , \tag{2.14}$$

where the *equivariant parameters* $\epsilon_i$ are defined ambiguously, up to the shifts $\epsilon_i \sim \epsilon_i + \sum_a Q_i^a \lambda_a$ with arbitrary $\lambda_a \in \mathbb{C}$. Such transformations, given eq. (2.5), only shift the Hamiltonian $H_\epsilon$ by a constant. We define the equivariant volume of $X_t$ by

$$\mathcal{F}_\epsilon(t) = \int_{X_t} e^{\varpi_t + H_\epsilon} , \tag{2.15}$$

assuming the integral converges. It can be computed by the integral over the product of $\mathbb{C}^N$ and the Lie algebra of the torus $U(1)^r$

$$\mathcal{F}_\epsilon(t) = \int_{\mathbb{C}^N} dz d\bar{z} \int_{\mathbb{R}^r} \prod_{a=1}^{r} \frac{d\phi_a}{2\pi} \exp{-\left[ \epsilon_i p^i + i\phi_a(Q_i^a p^i - t^a) \right]} , \tag{2.16}$$

where the last factor on the right hand side of eq. (2.16) comes from the integral representation of the delta-function:

$$\delta(x) = \frac{1}{2\pi} \int_{-\infty}^{+\infty} d\phi \; e^{-\mathrm{i}x\phi} \,. \tag{2.17}$$

Integrating out the $z$ variables in eq. (2.16), we arrive at the contour integral

$$\mathcal{F}_\epsilon(t) = \int_{(\mathrm{i}\mathbb{R})^r} \prod_{a=1}^{r} \frac{d\phi_a}{2\pi\mathrm{i}} \frac{e^{t^a \phi_a}}{\prod_{i=1}^{N}\left(\epsilon_i + Q_i^a \phi_a\right)} \tag{2.18}$$

and if we close the contour $(\mathrm{i}\mathbb{R}^r) \to C$ so that the integral is convergent for our choice of $t$, then we can calculate it by residues.

In simple cases one can determine the contour $C$ explicitly by requiring convergence of the above integral. However, with more integrations this becomes increasingly complicated and one uses the Jeffrey-Kirwan (JK) prescription for choosing the relevant poles [34, 35]. We do not review here the general JK prescription for picking the poles, but we present some explicit examples below. Assuming the appropriate contour $C$ with JK prescription, we define the equivariant volume of $X$ by[3]

$$\mathcal{F}_\epsilon(t) := \oint_C \prod_{a=1}^{r} \frac{d\phi_a}{2\pi\mathrm{i}} \frac{e^{t^a \phi_a}}{\prod_{i=1}^{N}\left(\epsilon_i + Q_i^a \phi_a\right)} \,. \tag{2.19}$$

This satisfies the property

$$\mathcal{F}_{\epsilon + Q^a \lambda_a}(t) = e^{-t^a \lambda_a} \mathcal{F}_\epsilon(t), \tag{2.20}$$

which may allow to set some $\epsilon$'s (or their combinations) to zero, and the scaling property

$$\mathcal{F}_{\lambda\epsilon}(\lambda^{-1}t) = \lambda^{-d} \mathcal{F}_\epsilon(t). \tag{2.21}$$

If $X$ is compact, then the above integral is regular in the equivariant parameters $\epsilon$'s around $\epsilon_i = 0$ for all $i = 1, \ldots, N$ (i.e., all $\epsilon$'s can be set to zero) and at zeroth order in $\epsilon$'s it gives the volume of Delzant polytope (up to irrelevant numerical factors), or equivalently the symplectic volume of $X_t$. Thus for compact $X$, we have

$$\mathcal{F}_\epsilon(t) = \mathbf{p}_d(t) + O(\epsilon), \tag{2.22}$$

with $\mathbf{p}_d(t)$ a homogeneous polynomial of degree $d$. This polynomial has a clear topological meaning as it corresponds (see eq. (2.16)) to the intersection polynomial on $H^2(X, \mathbb{R})$, with $t$ being coordinates on $H^2$. Thus, eq. (2.19) is an efficient way to calculate it.

If $X$ is non-compact, which is the case we are mainly interested in, then $\mathcal{F}_\epsilon(t)$ has poles in $\epsilon$'s. The scaling property eq. (2.21) together with the definition eq. (2.19) gives

$$\mathcal{F}_\epsilon(t) = \frac{1}{R_d(\epsilon)} \left(1 + \sum_{i=1}^{\infty} P_i(t,\epsilon)\right), \tag{2.23}$$

where $R_d$ and $P_i$ are rational functions in $\epsilon$ and polynomials in $t$, satisfying

$$R_d(\lambda\epsilon) = \lambda^d R_d(\epsilon), \quad P_i(t, \lambda\epsilon) = \lambda^i P_i(t, \epsilon), \quad P_i(\lambda^{-1}t, \lambda\epsilon) = P_i(t, \epsilon). \tag{2.24}$$

In the literature the main focus was on the most singular term $R_d^{-1}(\epsilon)$, which is sometimes called equivariant volume. The less singular terms $P_i(t, \epsilon)R_d^{-1}(\epsilon)$ have not been studied systematically.

---

[3]When there's no ambiguity, we drop the dependence on the geometry $X$, by writing $\mathcal{F}$ instead of $\mathcal{F}^X$. We denote its dependence on equivariant parameters $\epsilon_i$ by the subscript index.

For example, an important question is whether one can extract from $P_d(\boldsymbol{t}, \epsilon)R_d^{-1}(\epsilon)$ a polynomial in $\boldsymbol{t}$ with clear geometric meaning, as for the compact case. Since we cannot set $\epsilon = 0$, there is no natural way to convert $P_d(\boldsymbol{t}, \epsilon)R_d^{-1}(\epsilon)$ to a polynomial in $\boldsymbol{t}$. In some special cases, upon a clever choice of $\epsilon$'s (using eq. (2.20)), $P_d(\boldsymbol{t}, \epsilon)R_d^{-1}(\epsilon)$ collapses to a polynomial in $\boldsymbol{t}$ only. In other examples this does not work, and there is always a dependence on $\epsilon$'s in $P_d(\boldsymbol{t}, \epsilon)R_d^{-1}(\epsilon)$. It is a goal of this work to explain the structure of $P_d(\boldsymbol{t}, \epsilon)R_d^{-1}(\epsilon)$ and their geometric meaning.

# 3 Quantum mechanical setup

As we said above, the Kähler quotient $X = \mathbb{C}^N // \mathbb{T}$ has another interesting description. On $\mathbb{C}^N$, consider the $\lambda \in \mathbb{T}_{\mathbb{C}}$ action $\boldsymbol{z} \mapsto \lambda \cdot \boldsymbol{z} := \left( z^i \prod_a \lambda_a^{Q_i^a} \right)_{i=1}^N$, with the same charge matrix $Q$ as before. Given a *polarization vector* $\boldsymbol{T} \in \mathbb{Z}^r$, let us define the open subset $\left( \mathbb{C}^N \right)^{\text{stable}} \subset \mathbb{C}^N$ of $\boldsymbol{T}$-stable points: a point $\boldsymbol{z}_*$ is *stable* if there exists a non-constant polynomial $\Psi(\boldsymbol{z})$ obeying the *Gauss law*:

$$\Psi(\lambda \cdot \boldsymbol{z}) = \Psi(\boldsymbol{z}) \prod_{a=1}^r \lambda_a^{T^a}, \tag{3.1}$$

which does not vanish at $\boldsymbol{z}_*$: $\Psi(\boldsymbol{z}_*) \neq 0$. Then, the theorem states:

$$X_{\boldsymbol{t}} = \left( \mathbb{C}^N \right)^{\text{stable}} / \mathbb{T}_{\mathbb{C}} \approx \mu^{-1}(\boldsymbol{t})/\mathbb{T}, \tag{3.2}$$

where $\boldsymbol{t} = \hbar \boldsymbol{T}$ for any positive $\hbar \in \mathbb{R}_+$. The symplectic structure depends on $\hbar$, while the complex structure does not.

More precisely, for integers $n^i \geq 0$ satisfying $Q_i^a n^i = T^a$, consider monomials

$$\Psi_n(\boldsymbol{z}) = \prod_i (z^i)^{n^i}. \tag{3.3}$$

If we choose $t^a = \hbar T^a$, then $X = \mu^{-1}(\boldsymbol{t})/\mathbb{T}$ gets equipped with an integral Kähler form $\varpi_{\boldsymbol{t}}/\hbar$. We denote by $\mathcal{L}_{\boldsymbol{T}}$ the associated line bundle on $X$, the monomials in eq. (3.3) are its holomorphic sections. They give rise to a basis that is diagonal under the torus $H = U(1)^N/\mathbb{T} \approx U(1)^{N-r}$-action on $X_{\boldsymbol{t}}$. One can define the norm $||\Psi(z)||^2$ on $X$ by choosing an appropriate Hermitian structure on $\mathcal{L}_{\boldsymbol{T}}$ (see appendix A for details) and construct the Hilbert space $\mathcal{H}_{\boldsymbol{T}}$.

Define the equivariant partition function (twisted index, or a character) as

$$Z_q(\boldsymbol{T}) = \text{tr}_{\mathcal{H}_{\boldsymbol{T}}} q = \sum_{\boldsymbol{n} \in \mathbb{Z}_+^N,\ Q \cdot \boldsymbol{n} = \boldsymbol{T}} \prod_i q_i^{n^i}, \tag{3.4}$$

which computes the $H$-character of the space of holomorphic sections of $\mathcal{L}_{\boldsymbol{T}}$. It defines a meromorphic function of $q \in H_{\mathbb{C}}$ (the sum in eq. (3.4) has, typically, only a finite convergence radius). This corresponds to counting integer points inside a polyhedron.[4] For compact $X$, $Z_q(\boldsymbol{T})$ is a Laurent polynomial in $q$, so we set $q = 1$, giving us the dimension $Z_1(\boldsymbol{T}) = \dim \mathcal{H}_{\boldsymbol{T}}$ of the associated Hilbert space.

Using the representation of periodic delta function, we can rewrite eq. (3.4) as

$$Z_q(\boldsymbol{T}) = \int_{[-i\pi, i\pi]^r} \prod_{a=1}^r \frac{d(\phi_a \hbar)}{2\pi i} \frac{e^{T^a \phi_a \hbar}}{\prod_{i=1}^N \left( 1 - q_i e^{-\hbar Q_i^a \phi_a} \right)} \tag{3.5}$$

and to calculate this integral we deform the contour of integration appropriately. The logic is similar to the one in previous section and we apply the JK prescription to select the poles. The

---

[4]There exists a better basis for its description; we present explicit examples below.

crucial fact is that the contours of integration in eq. (2.19) and its quantum version eq. (3.5) are the same. It is natural to rewrite the above integral as

$$Z_q(T) = (-1)^r \oint_\Gamma \prod_{a=1}^r \frac{dw_a}{2\pi i w_a} \frac{\prod_{a=1}^r w_a^{-T^a}}{\prod_{i=1}^N \left(1 - q_i \prod_{a=1}^r w_a^{Q_i^a}\right)}, \tag{3.6}$$

with an appropriate choice of contour $\Gamma$ and $w_a = e^{-\hbar\phi_a}$. Similarly to eq. (2.20), we have

$$Z_{e^{Q_i^a \lambda_a} q_i}(T) = e^{T^a \lambda_a} Z_q(T). \tag{3.7}$$

We can rewrite eq. (3.5) as

$$Z_q(T) = \frac{1}{\hbar^d} \oint_C \prod_{a=1}^r \frac{d\phi_a}{2\pi i} \frac{e^{t^a \phi_a}}{\prod_{i=1}^N (\epsilon_i + Q_i^a \phi_a)} \prod_{i=1}^N G\left(\hbar(\epsilon_i + Q_i^a \phi_a)\right), \tag{3.8}$$

where $q_i = e^{-\hbar\epsilon_i}$. If we set $x_i = \epsilon_i + Q_i^a \phi_a$, the function

$$G(\hbar x) = \frac{\hbar x}{1 - e^{-\hbar x}} = 1 + \frac{\hbar x}{2} + \frac{(\hbar x)^2}{12} - \frac{(\hbar x)^4}{720} + \cdots \tag{3.9}$$

in eq. (3.8) corresponds to the equivariant Todd class of $TX$,

$$\prod_i G(\hbar x_i) = 1 + \hbar \frac{c_1}{2} + \hbar^2 \frac{c_2 + c_1^2}{12} + \hbar^3 \frac{c_1 c_2}{24} + \hbar^4 \frac{-c_1^4 + 4c_1^2 c_2 + c_1 c_3 + 3c_2^2 - c_4}{720} + \cdots, \tag{3.10}$$

where $c_i$ are the elementary symmetric functions of variables $x_i$, namely

$$c_1 = \sum_i x_i, \quad c_2 = \sum_{i<j} x_i x_j, \quad c_3 = \sum_{i<j<k} x_i x_j x_k, \tag{3.11}$$

which correspond to Chern classes. We obtain the semi-classical expansion [36]

$$Z_q(T) = \frac{1}{\hbar^d} \int_X e^{\omega + H}\left(1 + \hbar \frac{c_1}{2} + \hbar^2 \frac{c_2 + c_1^2}{12} + O(\hbar^3)\right), \tag{3.12}$$

which should be understood equivariantly on non-compact manifolds. If $X$ is compact, then one can evaluate the integral in eq. (3.8) and set $\epsilon$'s to zero. Since there are no singular terms in $\epsilon$'s, we keep only relevant terms in eq. (3.8)

$$Z_1(T) = \frac{1}{\hbar^d} \int_C \prod_{a=1}^r \frac{d\phi_a}{2\pi i} \frac{1}{\prod_{i=1}^N (\epsilon_i + Q_i^a \phi_a)} \left[ \frac{(\phi \cdot t)^d}{d!} + \frac{\hbar}{2} c_1 \frac{(\phi \cdot t)^{d-1}}{(d-1)!} \right.$$
$$\left. + \frac{\hbar^2}{12}(c_1^2 + c_2) \frac{(\phi \cdot t)^{d-2}}{(d-2)!} + \cdots \right], \tag{3.13}$$

where lower powers of $\phi \cdot t$ do not contribute due to compactness. The expansion above is in powers of $T^a = t^a/\hbar$, and $\hbar$ can be absorbed there. Thus the semi-classical expansion of $Z_q(T)$ can be understood as an expansion in powers of $T$, and large $T$ is the same as $\hbar^{-1}$. So

$$Z_1(T) = \sum_{i=1}^d \text{Pol}_i(T), \tag{3.14}$$

with the homogeneous polynomials satisfying $\text{Pol}_i(\lambda T) = \lambda^i \text{Pol}_i(T)$, and

$$\text{Pol}_d(T) = \mathbf{p}_d(T) = \mathcal{F}_0(T) \tag{3.15}$$

defined in eq. (2.22). This interpretation holds only for compact $X$, since $T$ control the size of the finite dimensional Hilbert space. This is well-known for compact spaces (see ref. [37] and references therein). For non-compact $X$, this expansion does not make sense, since the Hilbert space is infinite-dimensional, and we need full equivariance to make sense of $Z_q(T)$.

To get the index of Dirac operator, in eq. (3.8) we use

$$G(\hbar x) = \frac{\hbar x}{e^{\hbar x/2} - e^{-\hbar x/2}}, \qquad (3.16)$$

which corresponds to $\hat{A}$ genus. The two partition functions are related by (cf. eq. (2.8))

$$Z_q^{\hat{A}}(T) = \left( \prod_i q_i \right)^{1/2} Z_q^{\text{Todd}}(T - \frac{1}{2}(Q^1, \ldots, Q^r)). \qquad (3.17)$$

In what follows we only discuss partition functions built out of the Todd class.

## 4 Shift equations from compact support cohomology

For a non-compact manifold $X$, of real dimension $2d$, let us denote by $H^k(X)$ the standard de Rham $k$-th cohomology and by $H^k_{\text{comp}}(X)$ the $k$-th cohomology with compact support. We have the dualities $H^k(X) \approx H^{2d-k}_{\text{comp}}(X) \approx H_k(X)$, where $H_k(X)$ is the $k$-th homology group of $X$. The cohomology with compact support $H^k_{\text{comp}}(X)$ acts on $H^k(X)$

$$H^k(X) \times H^k_{\text{comp}}(X) \to H^k(X) \qquad (4.1)$$

by the shifts: $(\omega, \omega_{\text{comp}}) \mapsto \omega + \omega_{\text{comp}}$. Moreover there exists a natural map

$$H^k_{\text{comp}}(X) \to H^k(X), \qquad (4.2)$$

which is defined via Poincaré duality for compact cycles. For details see the book [38].

Let us specialize to toric non-compact $X$. We assume that $H^2_{\text{comp}}(X)$ is non-empty (for the empty case, see section 6). We want to describe the maps eqs. (4.1) and (4.2) more concretely. On non-compact $X$ with Kähler form $\omega$, we choose a basis for $H_2(X)$

$$t^a = \int_{C^a} \omega, \qquad (4.3)$$

where $C^a$ is a basis in $H_2(X, \mathbb{Z})$. For the actual Kähler form $\omega$, the $t$ should satisfy certain inequalities. Let us denote by $D_i$ the toric divisors and by $\text{PD}\, D_i$ their Poincaré dual classes, represented by closed two-forms. We can write the charge matrix as

$$\int_{C^a} \text{PD}\, D_i = Q_i^a. \qquad (4.4)$$

Let us denote by $I$ those $i$ corresponding to compact toric divisors.[5] The Poincaré duals $\omega_I = \text{PD}\, D_I$ of compact toric divisors $D_I$ generate $H^2_{\text{comp}}$. Introduce $m^I$ as expansion coefficients for a compactly supported two-form $\omega_{\text{comp}} = -m^I \omega_I$ (here the minus is for later convenience), so

$$\int_{C^a} (\omega - m^I \omega_I) = t^a - Q_I^a m^I \qquad (4.5)$$

describes the map $H^2 \times H^2_{\text{comp}} \to H^2$. We call it the $\psi$-map, $t \mapsto t + \psi \cdot m$, given by

$$\psi = -(Q_I^a) : H^2_{\text{comp}}(X) \to H^2(X), \qquad (4.6)$$

which is a realization of the natural map eq. (4.2) for second cohomology groups.

---

[5]Explicitly, $\sum_I m_I \epsilon^I = m_I \epsilon^I = \sum_{i \mid D_i \text{ compact}} m_i \epsilon^i$.

## 4.1 Shift equation

Our goal is to understand how the equivariant volume $\mathcal{F}_\epsilon(t)$ defined in eq. (2.19) behaves under the action of compact support cohomology. The main statement of this paper is the following *shift equation*

$$\mathcal{F}_\epsilon(t) - e^{-m \cdot \epsilon} \mathcal{F}_\epsilon(t + \psi \cdot m) = \wp_d(t, m) + O(\epsilon) = \text{regular in } \epsilon, \tag{4.7}$$

where $m$'s correspond to compact divisors and we denote $m \cdot \epsilon = \sum_I m^I \epsilon_I$, where the sum runs over compact divisors. The non-trivial content of this equation is that RHS is regular in $\epsilon$'s, and at zeroth order in $\epsilon$'s we have a homogeneous polynomial $\wp_d(t, m)$ of degree $d$, $\wp_d(\lambda t, \lambda m) = \lambda^d \wp_d(t, m)$. The definition of $\mathcal{F}_\epsilon(t)$ requires the specification of a chamber in $t$, thus if we require that $t + \psi \cdot m$ are in the same chamber, then $\wp_d(t, m)$ is a concrete homogeneous polynomial. If we change chamber, then this polynomial may change as well. Equation (4.7) imposes constraints on the expansion eq. (2.23): the terms $P_i(t, \epsilon) R_d^{-1}(\epsilon)$ for $i < d$ depend on specific combinations of $t$ ($\dim H^2 - \dim H^2_{\text{comp}}$), as we explain below.

To prove eq. (4.7), consider a compact toric divisor $D_I$, defined by the equation $p^I = 0$, in addition to the general moment map equations. By definition of compact divisor, the space of $p$'s satisfying these equations is compact. Equivalently, if we remove the $I$-th column from the charge matrix $(Q_i^a)$, then we obtain the charge matrix of a compact space. Consider

$$\mathcal{F}_\epsilon(t) - e^{-m^I \epsilon_I} \mathcal{F}_\epsilon(t + \psi \cdot m) = \oint_C \prod_{a=1}^r \frac{d\phi_a}{2\pi i} \frac{e^{t^a \phi_a}}{\prod_{i=1}^N \left(\epsilon_i + Q_i^a \phi_a\right)} (1 - e^{-m^I(\epsilon_I + Q_I^a \phi_a)}) \tag{4.8}$$

and observe that

$$\frac{1 - e^{-m^I(\epsilon_I + Q_I^a \phi_a)}}{\prod_{i=1}^N \left(\epsilon_i + Q_i^a \phi_a\right)} = \sum_I \frac{\cdots}{\prod_{i=1, i \neq I}^N \left(\epsilon_i + Q_i^a \phi_a\right)}, \tag{4.9}$$

where dots contain (infinitely many) polynomial terms in $\phi$, $\epsilon$, and $m$ (starting from the linear term in $m$), while at denominator we have the $Q$-matrix of a *compact* space. We conclude that eq. (4.8) is regular in $\epsilon$'s and its zeroth order corresponds to a homogeneous polynomial.

Let us elaborate on the geometry behind our manipulations. We define correlators for toric divisors $(D_{j_1}, D_{j_2}, \ldots, D_{j_k})$

$$\langle D_{j_1} D_{j_2} \cdots D_{j_k} \rangle_\epsilon = \prod_{s=1}^k \left( \epsilon_{i_s} + Q_{i_s}^b \frac{\partial}{\partial t^b} \right) \mathcal{F}_\epsilon(t). \tag{4.10}$$

These correlators correspond to equivariant volumes of divisors, their intersections, etc. For example, for one divisor we have

$$\langle D_i \rangle_\epsilon = \text{vol}_\epsilon(D_i) \tag{4.11}$$

and for two divisors

$$\langle D_i D_j \rangle_\epsilon = \text{vol}_\epsilon(D_i \cap D_j). \tag{4.12}$$

These correlators contain singular terms in $\epsilon$ unless at least one of the divisors is compact:

$$\langle D_I D_{j_2} \cdots D_{j_k} \rangle_\epsilon = \text{regular in } \epsilon, \tag{4.13}$$

where $D_I$ is a compact divisor. Thus we can calculate intersection numbers from DH theorem only when at least one of the divisors is compact

$$\langle D_I D_{j_2} \cdots D_{j_d} \rangle_\epsilon = c_{I j_2 \ldots j_d} + O(\epsilon), \tag{4.14}$$

where $d = \dim_{\mathbb{C}} X$. The analogous object with non-compact divisors only always contains singular terms in $\epsilon$ and there is no natural way to associate intersection numbers to these divisors within this framework. Using this terminology the shift eq. (4.8) becomes

$$\langle 1 - e^{-m^I D_I} \rangle_\epsilon = \wp_d(\boldsymbol{t}, m) + O(\epsilon), \tag{4.15}$$

where $I$ runs over all compact divisors. Thus the polynomial $\wp_d(\boldsymbol{t}, m)$ has the interpretation

$$\wp_d(\boldsymbol{t}, m) = m^I \operatorname{vol}_0(D_I) - \frac{1}{2} m^I m^J \operatorname{vol}_0(D_I \cap D_J) + \cdots. \tag{4.16}$$

We instead keep all equivariant parameters in place and avoid ad hoc choices, e.g. postulating a particular answer for the intersection number of non-compact divisors. As we explain below, we can define intersection numbers for all divisors upon certain non-canonical choices.

Equation (4.7) has a quantum analog, which can be proven in a similar fashion. Take integer $T$'s, which belong to some chamber, and define $Z_q(\boldsymbol{T})$ as in eq. (3.6). If we take the collection of integers $M^I$ (with $I$ being a label for compact divisors), then we have the following quantum shift equation

$$Z_q(\boldsymbol{T}) - Z_q(\boldsymbol{T} + \psi \cdot M) \prod_I q_I^{M^I} = \mathcal{P}(\boldsymbol{T}, M; q) = \text{a polynomial in } q, \tag{4.17}$$

where we assume that $\boldsymbol{T} + \psi \cdot M$ are in the same chamber as $\boldsymbol{T}$, in order for $\mathcal{P}(\boldsymbol{T}, M; q)$ to be a well-defined polynomial. It is crucial that $\mathcal{P}(\boldsymbol{T}, M; q)$ is an integer-valued polynomial, namely it belongs to $\mathbb{Z}[q_1, \ldots, q_N]$ for any admissible choice of $\boldsymbol{T}, M$. In examples below we illustrate some chamber structure issues for $\mathcal{P}$. Since the pole structure of $\mathcal{F}_\epsilon(\boldsymbol{t})$ and $Z_q(\boldsymbol{T})$ are the same, the proof of the quantum shift equation is analogous to the classical case.

Let us discuss the semi-classical expansion of the quantum shift equation. Since $\mathcal{P}(\boldsymbol{T}, M; q)$ is a polynomial in $q$'s, we can set $q = 1$ and obtain another polynomial in $\boldsymbol{T}$'s and $M$'s

$$\mathcal{P}(\boldsymbol{T}, M; 1) = \sum_{i=1}^{d} \mathcal{P}_i(\boldsymbol{T}, M), \tag{4.18}$$

where $\mathcal{P}_i(\lambda \boldsymbol{T}, \lambda M) = \lambda^i \mathcal{P}_i(\boldsymbol{T}, M)$. The leading term corresponds to

$$\mathcal{P}_d(\boldsymbol{T}, M) = \wp_d(\boldsymbol{T}, M) = \hbar^{-d} \wp_d(\boldsymbol{t}, m), \tag{4.19}$$

with $\wp_d$ being the classical polynomial in eq. (4.7) and $\boldsymbol{t} = \hbar \boldsymbol{T}$, $m = \hbar M$. In general, we have the semi-classical expansion

$$\mathcal{P}(\boldsymbol{T}, M; 1) = \hbar^{-d} \wp_d(\boldsymbol{t}, m) + \hbar^{-d+1} \mathcal{P}_{d-1}(\boldsymbol{t}, m) + \hbar^{-d+2} \mathcal{P}_{d-2}(\boldsymbol{t}, m) + \cdots, \tag{4.20}$$

where the correction $\mathcal{P}_{d-1}(\boldsymbol{t}, m)$ is controlled by $c_1$, $\mathcal{P}_{d-2}(\boldsymbol{t}, m)$ by $c_1^2 + c_2$ etc. Thus we can also write the shift equation order by order in $\hbar$.

Let us conclude with an observation. If we pick a compact divisor $CD$ with equivariant parameter $q_{CD}$, then formally we can write the partition function as a single infinite sum

$$Z_q(\boldsymbol{T}) = \sum_{n=0}^{\infty} q_{CD}^n Z_q^{CD}(\boldsymbol{T} + \psi \cdot n), \tag{4.21}$$

with $Z_q^{CD}(\boldsymbol{T})$ the partition function for a compact divisor (a finite sum). As $\boldsymbol{T} + \psi \cdot n$ may be outside of the chamber for $\boldsymbol{T}$, we have to replace $Z_q^{CD}(\boldsymbol{T} + \psi \cdot n)$ with different polynomials for different parts of the sum over $n$. Although it is hard to manipulate, eq. (4.21) provides a nice physical intuition behind our quantum shift equation.

## 4.2 Solving the shift equation

Let us discuss how to solve eq. (4.7). From one side, this is an algebraic equation, but we have to keep in mind the chamber structure in $t$. Let us assume that $t$ and $t + \psi \cdot m$ belong to the same chamber (they can also lie on the boundary of the chamber).

Let us start from algebraic aspects. We identified the map $\psi : \mathbb{R}^b \to \mathbb{R}^r$ with certain components of the charge matrix, where $b = \dim H^2_{\text{comp}}(X)$ is the number of compact toric divisors. Let's choose a basis $t^\alpha = O^\alpha_a t^a$, with $\alpha = 1, \ldots, r - b$ for $\text{coker}\,\psi$, and a basis $t^I$, with $I = 1, \ldots, b$ for $\text{im}\,\psi$, where the matrix $\psi$ is invertible. This choice is not canonical. If we choose $m$'s such that

$$Q^J_I m^I = t^J \tag{4.22}$$

then the second term in LHS of eq. (4.7) becomes

$$e^{-m \cdot \epsilon} \mathcal{F}_\epsilon(t + \psi \cdot m) = e^{-\epsilon_J (Q^{-1})^J_I t^I} \mathcal{F}_\epsilon(t_\alpha, 0). \tag{4.23}$$

For the sake of clarity, using eq. (2.20) we can set $\epsilon_I = 0$ and get

$$\mathcal{F}_\epsilon(t) = \mathcal{F}_\epsilon(t^\alpha, 0) + \wp_d(t^\alpha, (Q^{-1})^I_J t^J) + O(\epsilon). \tag{4.24}$$

This is a solution to the shift equation, which depends on the choice of bases above. As far as algebra considerations, all bases are equivalent. However, if we impose a chamber structure, this is not true anymore. The above solution is valid if the point $(t^\alpha, 0) = (O^\alpha_a t^a, 0)$ lays in the same chamber (or its boundary) as the original $t$.

Another way to interpret this solution is to say that we compare the equivariant volumes of two spaces: one $\mathcal{F}_\epsilon(t)$ for the regular $X$ and another $\mathcal{F}_\epsilon(t^\alpha, 0)$ for the singular relative of $X$, where we degenerate some $t$ in a particular way (staying in the same chamber or its boundary). By comparing these two equivariant volumes, we obtain the regular term

$$\mathcal{F}_\epsilon(t) - \mathcal{F}_\epsilon(t^\alpha, 0) = \wp_d(t^\alpha, (Q^{-1})^I_J t^J) + O(\epsilon) \tag{4.25}$$

and we can interpret the polynomial $\wp_d$ as a generalized intersection for non-compact spaces, which depends on the choices above. In this representation we see that the singular terms in $\mathcal{F}_\epsilon(t)$ depend only on $t^\alpha$.

# 5 Examples with nontrivial $H^2$ with compact support

We present explicit examples of shift equations for cases with non-empty $H^2_{\text{comp}}(X)$. These examples illustrate and clarify our general discussion. We try to be explicit and emphasize the different possible relations between $H^2(X)$ and $H^2_{\text{comp}}(X)$. To avoid cluttering notation, in examples we use lower indices for both $t$ and $p$.

## 5.1 $A_1$-space

We realize the total space $X = \text{Tot}(\mathcal{O}(-2) \to \mathbb{P}^1)$ as the quotient of $\mathbb{C}^3$ by $U(1)$ with

$$Q = \begin{pmatrix} 1 & -2 & 1 \end{pmatrix}. \tag{5.1}$$

According to eq. (2.4), momenta satisfy

$$p_1 - 2p_2 + p_3 = t. \tag{5.2}$$

This space has one-dimensional $H^2$ and one-dimensional $H^2_{\text{comp}}$, since it has one compact divisor, given by $X \cap \{p_2 = 0\}$. The $\psi$-map is non-degenerate in this case. The equivariant volume is

$$\mathcal{F}_{\epsilon}(t) = \oint_C \frac{d\phi}{2\pi i} \frac{e^{\phi t}}{(\epsilon_1 + \phi)(\epsilon_2 - 2\phi)(\epsilon_3 + \phi)}, \tag{5.3}$$

where only poles at $-\epsilon_1$ and $-\epsilon_3$ contribute. For this space eq. (4.7) becomes

$$\mathcal{F}_{\epsilon}(t) - e^{-m\epsilon_2} \mathcal{F}_{\epsilon}(t + 2m) = mt + m^2 + O(\epsilon) \tag{5.4}$$

and since there is no kernel for $\psi$-map the solution is

$$\mathcal{F}_{\epsilon}(t) = e^{t\epsilon_2/2} \mathcal{F}_{\epsilon}(0) - \frac{1}{4}t^2 + O(\epsilon). \tag{5.5}$$

The partition function

$$Z_q(T) = \sum_{s_1, s_2, s_3 \geq 0, \ s_1 - 2s_2 + s_3 = T} q_1^{s_1} q_2^{s_2} q_3^{s_3} \tag{5.6}$$

satisfies the quantum shift equation eq. (4.17)

$$Z_q(T) - q_2^M Z_q(T + 2M) = \mathcal{P}(T, M; q) = \sum_{l=0}^{M-1} \sum_{s=0}^{T+2l} q_1^s q_2^l q_3^{T+2l-s}, \tag{5.7}$$

where we assume $M$ to be a positive integer. If we evaluate this polynomial at $q = 1$, we get

$$\mathcal{P}(T, M; 1) = TM + M^2, \tag{5.8}$$

which agrees with the RHS of eq. (5.4) (no quantum corrections).

Since $X$ is the total space of a line bundle, the partition function can be expanded in terms of compact divisor partition functions

$$Z_q(T) = \sum_{n=0}^{\infty} q_2^n \sum_{s=0}^{T+2n} q_1^s q_3^{T+2n-s} = \sum_{n=0}^{\infty} q_2^n Z_{q_1,q_3}^{\mathbb{P}^1}(T + 2n). \tag{5.9}$$

However, such simple expressions are not available in more complicated examples, for example for the next example of $A_2$-space.

## 5.2 $A_2$-space

Consider a four-dimensional $X$, constructed as the quotient of $\mathbb{C}^4$ by $U(1)^2$ with

$$Q = \begin{pmatrix} 1 & -2 & 1 & 0 \\ 0 & 1 & -2 & 1 \end{pmatrix}. \tag{5.10}$$

Equation (2.4) becomes

$$\begin{aligned} p_1 - 2p_2 + p_3 &= t_1, \\ p_2 - 2p_3 + p_4 &= t_2. \end{aligned} \tag{5.11}$$

This space has a two-dimensional $H^2$ and two-dimensional $H^2_{\text{comp}}$, with the two compact toric divisors given by $X \cap \{p_2 = 0\}$ and $X \cap \{p_3 = 0\}$. The equivariant volume is

$$\mathcal{F}_{\epsilon}(t) = \oint_C \frac{d\phi_1 d\phi_2}{(2\pi i)^2} \frac{e^{\phi_1 t_1} e^{\phi_2 t_2}}{(\epsilon_1 + \phi_1)(\epsilon_2 - 2\phi_1 + \phi_2)(\epsilon_3 + \phi_1 - 2\phi_2)(\epsilon_4 + \phi_2)}, \tag{5.12}$$

where poles at $(-\epsilon_1, -\epsilon_2 - 2\epsilon_1)$, $(-\epsilon_1, -\epsilon_4)$ and $(-\epsilon_3 - 2\epsilon_4, -\epsilon_4)$ contribute. We get

$$\mathcal{F}_\epsilon(t) = -\frac{e^{-\epsilon_1 t_1 - (2\epsilon_1 + \epsilon_2)t_2}}{(3\epsilon_1 + 2\epsilon_2 + \epsilon_3)(2\epsilon_1 + \epsilon_2 - \epsilon_4)} - \frac{e^{-\epsilon_1 t_1 - \epsilon_4 t_2}}{(\epsilon_1 - \epsilon_3 - 2\epsilon_4)(2\epsilon_1 + \epsilon_2 - \epsilon_4)}$$
$$-\frac{e^{-(\epsilon_3 + 2\epsilon_4)t_1 - \epsilon_4 t_2}}{(-\epsilon_1 + \epsilon_3 + 2\epsilon_4)(\epsilon_2 + 2\epsilon_3 + 3\epsilon_4)} \quad (5.13)$$

and we can derive the shift equation

$$\mathcal{F}_\epsilon(t_1, t_2) - e^{-m_1 \epsilon_2 - m_2 \epsilon_3}\mathcal{F}_\epsilon(t_1 + 2m_1 - m_2, t_2 - m_1 + 2m_2)$$
$$= t_1 m_1 + t_2 m_2 + m_1^2 + m_2^2 - m_1 m_2 + O(\epsilon), \quad (5.14)$$

where we assume the chamber $t_1 \geq 0$, $t_2 \geq 0$ and $t_1 + 2m_1 - m_2 \geq 0$, $t_2 - m_1 + 2m_2 \geq 0$.

In this example the $\psi$-map is non-degenerate and we can solve this shift equation by

$$\mathcal{F}_\epsilon(t) = e^{(2\epsilon_2 + \epsilon_3)t_1/3 + (\epsilon_2 + 2\epsilon_3)t_2/3}\mathcal{F}_\epsilon(0,0) - \frac{1}{3}(t_1^2 + t_1 t_2 + t_2^2) + O(\epsilon) \quad (5.15)$$

if we choose $m_1 = -\frac{1}{3}(t_2 + 2t_1)$, $m_2 = -\frac{1}{3}(t_1 + 2t_2)$, which is on the boundary of the above chamber. The partition function

$$Z_q(T) = \sum_{s_1, s_2, s_3, s_4 \geq 0, \, s_1 - 2s_2 + s_3 = T_1, \, s_2 - 2s_3 + s_4 = T_2} q_1^{s_1} q_2^{s_2} q_3^{s_3} q_4^{s_4} \quad (5.16)$$

satisfies the quantum shift equation

$$Z_q(T) - q_2^{M_1} q_3^{M_2} Z_q(T_1 + 2M_1 - M_2, T_2 - M_1 + 2M_2) = \mathcal{P}(T, M; q), \quad (5.17)$$

with $\mathcal{P}$ a polynomial in $q$'s, whose concrete form depends on the values of $T$ and $M$'s. In the region where $T_1 + 2M_1 - M_2 \geq 0$, $T_2 + 2M_2 - M_1 \geq 0$ and $M_1, M_2 \geq 0$, we have

$$\mathcal{P}(T, M; q) = q_1^{T_1} q_4^{T_2} \sum_{(s_2, s_3) \in R} (q_1^2 q_2 q_4^{-1})^{s_2} (q_3 q_4^2 q_1^{-1})^{s_3}, \quad (5.18)$$

where the region of summation

$$R = \{(s_2, s_3) \in \mathbb{Z}^2 \mid s_2, s_3 \geq 0; \, s_3 - 2s_2 \leq T_1; \, s_2 - 2s_3 \leq T_2; \, s_2 < M_1 \text{ or } s_3 < M_2\} \quad (5.19)$$

has size

$$\mathcal{P}(T, M; 1) = \sum_{(s_2, s_3) \in R} 1 = M_1 T_1 + M_1^2 + M_2 T_2 + M_2^2 - M_1 M_2, \quad (5.20)$$

which agrees with the RHS of eq. (5.14) without quantum corrections. If we further assume $T_2 - M_1 + 1 \geq 0$, $T_1 + 2M_1 - M_2 + 1 \geq 0$ and $M_1, M_2 \geq 1$, we get

$$\mathcal{P}(T, M; q) = \sum_{s=0}^{M_2 - 1} \sum_{k=0}^{T_2 - M_1 + 2s} q_1^{3s + T_1 + 2T_2 - 2k} q_2^{2s + T_2 - k} q_3^s q_4^k$$
$$+ \sum_{s=0}^{M_1 - 1} \sum_{k=0}^{T_1 + 2s} q_1^k q_2^s q_3^{2s + T_1 - k} q_4^{3s + 2T_1 + T_2 - 2k} \quad (5.21)$$

and its evaluation at $q = 1$ agrees with eq. (5.20).

Let us comment on the analog of eq. (4.21) for this space. Formally, we can write the expansion in terms of $q_2$ (which controls the compact divisor $p_2 = 0$)

$$Z_q(T) = \sum_{n=0}^{\infty} q_2^n Z_q^{CD}(T_1 + 2n, T_2 - n), \quad (5.22)$$

with $Z_q^{CD}$ being a finite sum. However, $Z_q^{CD}$ is a different function depending on the value of $T_2 - n$. For $T_2 \geq n$, the polynomial $Z_q^{CD}$ is the partition function of a compact divisor.

### 5.3 $SU(3)$ example

Consider the Calabi-Yau three-fold given by quotient of $\mathbb{C}^6$ by $U(1)^3$ with

$$Q = \begin{pmatrix} 1 & 1 & 1 & -3 & 0 & 0 \\ 0 & 0 & 1 & -2 & 1 & 0 \\ 0 & 0 & 0 & 1 & -2 & 1 \end{pmatrix}. \tag{5.23}$$

Momenta satisfy

$$\begin{aligned} p_1 + p_2 + p_3 - 3p_4 &= t_1, \\ p_3 - 2p_4 + p_5 &= t_2, \\ p_4 - 2p_5 + p_6 &= t_3, \end{aligned} \tag{5.24}$$

where we assume $t_1 - t_2 > 0$, besides $t_a \geq 0$ for all $a$. This example was considered in ref. [19], where it engineers $SU(3)$ gauge theory with zero Chern-Simons level. Here we clarify a few points. For this space, $H^2$ is three dimensional while $H^2_{\text{comp}}$ is two dimensional, with the corresponding compact toric divisors given by $X \cap \{p_4 = 0\}$ and $X \cap \{p_5 = 0\}$. Thus this is the first example when our $\psi$-map has a one-dimensional co-kernel. The shift equation reads

$$\mathcal{F}_\epsilon(\boldsymbol{t}) - e^{-\epsilon_4 m_1 - \epsilon_5 m_2} \mathcal{F}_\epsilon(t_1 + 3m_1, t_2 + 2m_1 - m_2, t_3 - m_1 + 2m_2) =$$
$$\frac{4}{3}m_1^3 + \frac{4}{3}m_2^3 - \frac{1}{2}m_1^2 m_2 - \frac{1}{2}m_1 m_2^2$$
$$+ m_1^2(t_1 + \frac{1}{2}t_2) + m_2^2(t_1 - t_2 + \frac{3}{2}t_3) + m_1 m_2(t_2 - t_1)$$
$$+ m_1(t_1 t_2 - \frac{1}{2}t_2^2) + m_2(t_1 t_3 - t_2 t_3 + \frac{1}{2}t_3^2) + O(\epsilon), \tag{5.25}$$

where we require $t_1 - t_2 > -m_1 - m_2$ and $t_3 - m_1 + 2m_2 > 0$ to remain in the same chamber. We can approach the boundary in two ways: either by setting $m_1 = -\frac{1}{3}t_1$ and $m_2 = t_2 - \frac{2}{3}t_1$, in the sub-chamber $t_3 + 2t_2 - t_1 > 0$

$$\mathcal{F}_\epsilon(t_1, t_2, t_3) = e^{\epsilon_4 t_1/3 + \epsilon_5(2t_1 - 3t_2)/3} \mathcal{F}_\epsilon(0, 0, t_3 + 2t_2 - t_1)$$
$$+ \frac{t_2^3}{3} - \frac{t_1 t_2^2}{3} - \frac{t_1 t_2 t_3}{3} + \frac{t_2^2 t_3}{2} - \frac{t_1 t_3^2}{3} + \frac{t_2 t_3^2}{2} + O(\epsilon) \tag{5.26}$$

or by setting $m_1 = -\frac{1}{3}t_3 - \frac{2}{3}t_2$ and $m_2 = -\frac{2}{3}t_3 - \frac{1}{3}t_2$, in the sub-chamber $t_3 + 2t_2 - t_1 < 0$

$$\mathcal{F}_\epsilon(t_1, t_2, t_3) = e^{\epsilon_4(t_3 + 2t_2)/3 + \epsilon_5(2t_3 + t_2)/3} \mathcal{F}_\epsilon(t_1 - t_3 - 2t_2, 0, 0)$$
$$+ \frac{t_2^3}{3} - \frac{t_1 t_2^2}{3} - \frac{t_1 t_2 t_3}{3} + \frac{t_2^2 t_3}{2} - \frac{t_1 t_3^2}{3} + \frac{t_2 t_3^2}{2} + O(\epsilon), \tag{5.27}$$

which gives the same five dimensional prepotential $\wp_3(t, m(t))$. One can also set $m_1 = -\frac{1}{3}t_1$ and $m_2 = -\frac{1}{6}t_1 - \frac{1}{2}t_3$

$$\mathcal{F}_\epsilon(t_1, t_2, t_3) = e^{\epsilon_4 t_1/3 + \epsilon_5(t_1 + 3t_3)/6} \mathcal{F}_\epsilon(0, t_2 - \frac{1}{2}t_1 + \frac{1}{2}t_3, 0)$$
$$- \frac{1}{24}\left(-t_1^3 + t_3^2(-6t_2 + t_3) + 3t_1^2(2t_2 + t_3) + t_1(-4t_2^2 - 4t_2 t_3 + 5t_3^2)\right) + O(\epsilon), \tag{5.28}$$

with the caveat that this choice lies outside of the chamber. The partition function satisfies

$$Z_q(\boldsymbol{T}) - q_4^{M_1} q_5^{M_2} Z_q(T_1 + 3M_1, T_2 + 2M_1 - M_2, T_3 - M_1 + 2M_2) = \mathcal{P}(\boldsymbol{T}, M; q), \tag{5.29}$$

with the polynomial

$$\mathcal{P}(\boldsymbol{T}, M; 1) = \frac{4}{3}(M_1^3 + M_2^3) - \frac{1}{2}(M_1^2 M_2 + M_1 M_2^2)$$
$$+ M_1^2 (T_1 + \frac{1}{2} T_2) + M_2^2 (T_1 - T_2 + \frac{3}{2} T_3) + M_1 M_2 (T_2 - T_1)$$
$$+ M_1 (T_1 T_2 - \frac{1}{2} T_2^2) + M_2 (T_1 T_3 - T_2 T_3 + \frac{1}{2} T_3^2)$$
$$- \frac{1}{3}(M_1 + M_2). \quad (5.30)$$

Here we see quantum corrections to eq. (5.25), in the last line. In agreement with eq. (4.20), this has the expected form for a Calabi-Yau threefold, namely $\mathcal{P}(\boldsymbol{T}, M; 1) = \mathcal{P}_3(\boldsymbol{T}, M) + \mathcal{P}_1(\boldsymbol{T}, M)$, where the quantum correction $\mathcal{P}_1$ corresponds to genus-one constant map contribution for topological strings, provided we impose a relation between $m$ and $t$ as above.

Compare with the discussion around Fig. 5 of ref. [21].

## 5.4 Local del Pezzo 2

Consider the Calabi-Yau three-fold $X$ defined as symplectic quotient of $\mathbb{C}^6$ by $U(1)^3$, with

$$Q = \begin{pmatrix} 1 & 1 & 1 & 0 & 0 & -3 \\ 1 & 0 & 1 & -1 & 0 & -1 \\ 0 & 1 & 1 & 0 & -1 & -1 \end{pmatrix}. \quad (5.31)$$

According to eq. (2.4), momenta satisfy

$$p_1 + p_2 + p_3 - 3p_6 = t_1,$$
$$p_1 + p_3 - p_4 - p_6 = t_2, \quad (5.32)$$
$$p_2 + p_3 - p_5 - p_6 = t_3.$$

We assume $t_1 - t_2 - t_3 \geq 0$, $t_2 \geq 0$, and $t_3 \geq 0$. This space has $\dim H^2(X) = 3$ and $\dim H^2_{\text{comp}}(X) = 1$, the compact toric divisor being $X \cap \{p_6 = 0\}$. The space of $p$'s satisfying eq. (5.32) is a non-compact convex body, whose five vertices are located at

$$\begin{array}{c|l} v_1 & p_3 = t_1, \ p_4 = t_1 - t_2, \ p_5 = t_1 - t_3, \\ v_2 & p_2 = t_1 - t_2, \ p_3 = t_2, \ p_5 = t_1 - t_3, \\ v_3 & p_1 = t_1 - t_3, \ p_3 = t_3, \ p_4 = t_1 - t_2, \\ v_4 & p_1 = t_2, \ p_2 = t_1 - t_2, \ p_5 = t_1 - t_2 - t_3, \\ v_5 & p_1 = t_1 - t_3, \ p_2 = t_3, \ p_4 = t_1 - t_2 - t_3, \end{array} \quad (5.33)$$

with all remaining $p$'s set to zero. Vertices are connected by edges going from 1-3-5-4-2-1. The equivariant volume is computed by the residue

$$\mathcal{F}_\epsilon(t_1, t_2, t_3) = \oint_C \prod_{a=1}^{3} \frac{d\phi_a}{2\pi i}$$
$$\frac{e^{t_1 \phi_1 + t_2 \phi_2 + t_3 \phi_3}}{(\epsilon_1 + \phi_1 + \phi_2)(\epsilon_2 + \phi_1 + \phi_3)(\epsilon_3 + \phi_1 + \phi_2 + \phi_3)(\epsilon_4 - \phi_2)(\epsilon_5 - \phi_3)(\epsilon_6 - 3\phi_1 - \phi_2 - \phi_3)}, \quad (5.34)$$

where we take poles at the flags $(\phi_2, \phi_1, \phi_3) = (\epsilon_4, -\epsilon_1 - \epsilon_4, \epsilon_1 + \epsilon_4 - \epsilon_2)$, $(\phi_2, \phi_1, \phi_3) = (\epsilon_4, -\epsilon_1 - \epsilon_4, \epsilon_1 - \epsilon_3)$, $(\phi_3, \phi_1, \phi_2) = (\epsilon_5, -\epsilon_2 - \epsilon_5, \epsilon_2 + \epsilon_5 - \epsilon_1)$, $(\phi_3, \phi_1, \phi_2) = (\epsilon_5, -\epsilon_2 - \epsilon_5, \epsilon_2 - \epsilon_3)$, and $(\phi_2, \phi_3, \phi_1) = (\epsilon_4, \epsilon_5, -\epsilon_3 - \epsilon_4 - \epsilon_5)$, the last one with an extra minus sign. The shift equation reads

$$\mathcal{F}_\epsilon(\boldsymbol{t}) - e^{-m\epsilon_6}\mathcal{F}_\epsilon(t_1 + 3m, t_2 + m, t_3 + m) =$$
$$\frac{1}{2}\left(m(t_1^2 - t_2^2 - t_3^2) + m^2(3t_1 - t_2 - t_3) + \frac{7}{3}m^3\right) + O(\epsilon), \quad (5.35)$$

where we require $m \geq t_2 + t_3 - t_1$, $m \geq -t_2$ and $m \geq -t_3$. The quantum version reads

$$Z_q(\boldsymbol{T}) - q_6^M Z_q(T_1 + 3M, T_2 + M, T_3 + M) = \mathcal{P}(\boldsymbol{T}, M; q), \quad (5.36)$$

with the polynomial

$$\mathcal{P}(\boldsymbol{T}, M; 1) = \frac{7}{6}M^3 + \frac{1}{2}M^2(3T_1 - T_2 - T_3) + \frac{1}{2}M(T_1^2 - T_2^2 - T_3^2) - \frac{1}{6}M. \quad (5.37)$$

We see again a quantum correction in the last term.

## 5.5  $\mathcal{O}(-3, -3)$ over $\mathbb{P}^2 \times \mathbb{P}^2$

Consider the total space of the $\mathcal{O}(-3, -3)$ bundle over $\mathbb{P}^2 \times \mathbb{P}^2$. This Calabi-Yau five-fold $X$ is realized as the quotient of $\mathbb{C}^7$ by $U(1)^2$ with

$$Q = \begin{pmatrix} 1 & 1 & 1 & 0 & 0 & 0 & -3 \\ 0 & 0 & 0 & 1 & 1 & 1 & -3 \end{pmatrix}. \quad (5.38)$$

Momenta satisfy

$$p_1 + p_2 + p_3 - 3p_7 = t_1,$$
$$p_4 + p_5 + p_6 - 3p_7 = t_2. \quad (5.39)$$

This space has two-dimensional $H^2$ and one-dimensional $H^2_{\text{comp}}$, with the corresponding compact divisor given by $X \cap \{p_7 = 0\}$. The shift equation reads

$$\mathcal{F}_\epsilon(t_1, t_2) - e^{-m\epsilon_7}\mathcal{F}_\epsilon(t_1 + 3m, t_2 + 3m) =$$
$$\frac{81}{20}m^5 + \frac{27}{8}m^4(t_1 + t_2) + \frac{3}{4}m^3(t_2^2 + 4t_1 t_2 + t_1^2) + \frac{3}{4}m^2(t_1 t_2^2 + t_1^2 t_2) + \frac{1}{4}mt_1^2 t_2^2 + O(\epsilon),$$
$$(5.40)$$

where we assume $t_1 \geq 0$, $t_2 \geq 0$ and $t_1 + 3m \geq 0$, $t_2 + 3m \geq 0$. By choosing $m = -\frac{1}{3}t_1$, which lies on the boundary of the chamber, we obtain

$$\mathcal{F}_\epsilon(t_1, t_2) = e^{\epsilon_7 t_1/3}\mathcal{F}_\epsilon(0, t_2 - t_1) - \frac{t_1^5}{360} + \frac{t_1^4 t_2}{72} - \frac{t_1^3 t_2^2}{36}. \quad (5.41)$$

One can also produce a solution symmetric in $t_1$ and $t_2$, with the caveat that this choice of $t$'s does not lie in the same chamber as the one we started from.

We can also introduce the partition function

$$Z_q(T_1, T_2) = \oint \frac{dw_1 dw_2}{(2\pi i)^2 w_1 w_2}$$
$$\frac{w_1^{-T_1} w_2^{-T_2}}{(1 - q_1 w_1)(1 - q_2 w_1)(1 - q_3 w_1)(1 - q_4 w_2)(1 - q_5 w_2)(1 - q_6 w_2)(1 - q_7 w_1^{-3} w_2^{-3})} \quad (5.42)$$

and the quantum shift equation is

$$Z_q(T_1, T_2) - q_7^M Z_q(T_1 + 3M, T_2 + 3M) = \mathcal{P}(T_1, T_2, M; q). \quad (5.43)$$

For $M > 0$, the polynomial takes the form

$$\mathcal{P}(T_1, T_2, M; q) = \sum_{s=0}^{M-1} q_7^s \oint \frac{dw_1 dw_2}{(2\pi i)^2 w_1 w_2}$$
$$\frac{w_1^{-T_1-3s} w_2^{-T_2-3s}}{(1-q_1 w_1)(1-q_2 w_1)(1-q_3 w_1)(1-q_4 w_2)(1-q_5 w_2)(1-q_6 w_2)}. \quad (5.44)$$

A direct calculation gives

$$\mathcal{P}(T_1, T_2, M; q) = \sum_{s=0}^{M-1} q_7^s \sum_{l=0}^{T_1+3s} \sum_{m=0}^{T_1+3s-l} q_1^l q_2^m q_3^{T_1+3s-l-m} \sum_{p=0}^{T_2+3s} \sum_{k=0}^{T_2+3s-p} q_4^p q_5^k q_6^{T_2+3s-p-k}. \quad (5.45)$$

Using the elementary identities

$$\sum_{s=1}^{n} s = \frac{n(n+1)}{2}, \quad \sum_{s=1}^{n} s^2 = \frac{n(n+1)(2n+1)}{6},$$
$$\sum_{s=1}^{n} s^3 = \frac{n^2(n+1)^2}{4}, \quad \sum_{s=1}^{n} s^4 = \frac{n(n+1)(2n+1)(3n^2+3n-1)}{30} \quad (5.46)$$

we can calculate

$$\mathcal{P}(T_1, T_2, M; 1) = \sum_{s=0}^{M-1} \sum_{l=0}^{T_1+3s} \sum_{m=0}^{T_1+3s-l} \sum_{p=0}^{T_2+3s} \sum_{k=0}^{T_2+3s-p} 1$$
$$= \frac{1}{4} \sum_{s=0}^{M-1} \Big((T_1^2 + 3T_1 + 2) + 3s(2T_1 + 3) + 9s^2\Big)\Big((T_2^2 + 3T_2 + 2) + 3s(2T_2 + 3) + 9s^2\Big)$$
$$= \frac{81}{20} M^5 + \frac{27}{8} M^4(T_1 + T_2) + \frac{3}{4} M^3(T_1^2 + T_2^2 + 4T_1 T_2) + \frac{3}{4} M^2(T_1^2 T_2 + T_1 T_2^2) + \frac{1}{4} M T_1^2 T_2^2$$
$$- \frac{15}{4} M^3 - \frac{15}{8} M^2(T_1 + T_2) - \frac{1}{4} M(T_1^2 + T_2^2) - \frac{3}{4} M T_1 T_2$$
$$+ \frac{7}{10} M. \quad (5.47)$$

The different orders of this polynomial are controlled by the semi-classical expansion of the Todd class in eq. (3.10). The fifth-order line corresponds to the classical contribution in eq. (5.40). The forth-order is absent due to Calabi-Yau-condition ($c_1 = 0$). The third-order line is controlled by $c_2$ and the second-order line is absent due to Calabi-Yau-condition (since this term is controlled by $c_1 c_2$). The linear term in the last line is controlled by $3c_2^2 - c_4$.

# 6 Examples with higher compact cohomologies

In section 4 we derived the shift equations for non-empty $H_{comp}^2(X) = H_{2d-2}(X)$ and in section 5 we provided examples of that situation. There are cases when $H_{comp}^2(X)$ vanishes, but higher-degree cohomology with compact support does not, e.g. $H_{comp}^4(X) = H_{2d-4}(X) \neq 0$.

In this section, we concentrate on the case of non-zero $H_{comp}^4(X)$, assuming $H_{comp}^2(X) = 0$. We make a few observations, although a few aspects of this story are still unclear to us.

We inspect all intersections of two toric divisors $D_i$ and $D_j$, and those intersections that are compact generate $H_{comp}^4(X)$. Let us assume that the intersection of divisors $D_I$ and $D_J$ is

compact, for some *fixed I* and *J*. This means that, if from the charge matrix $(Q_i^a)$ we remove the $I$-th and $J$-th columns, then we obtain the charge matrix of a compact space. We can then repeat the procedure from section 4.1, but removing two columns in $Q$-matrix. We get

$$
\mathcal{F}_\epsilon(\boldsymbol{t}) - e^{-m^I \epsilon_I} \mathcal{F}_\epsilon(t^a - Q_I^a m^I) - e^{-m^J \epsilon_J} \mathcal{F}_\epsilon(t^a - Q_J^a m^J) + e^{-m^I \epsilon_I - m^J \epsilon_J} \mathcal{F}_\epsilon(t^a - Q_I^a m^I - Q_J^a m^J)
$$
$$
= \oint_C \prod_{a=1}^r \frac{d\phi_a}{2\pi i} \frac{e^{t^a \phi_a}(1 - e^{-m^I(\epsilon_I + Q_I^a \phi_a)})(1 - e^{-m^J(\epsilon_J + Q_J^a \phi_a)})}{\prod_{i=1}^N (\epsilon_i + Q_i^a \phi_a)}, \quad (6.1)
$$

where we do *not* sum over $I$ and $J$. We obtain regular expressions in $\epsilon$'s, since in denominators we effectively have the charge matrix of a compact space. The novelty is that we get a second-order difference equation. We can write

$$
\mathcal{F}_\epsilon(\boldsymbol{t}) - e^{-m \cdot \epsilon} \mathcal{F}_\epsilon(\boldsymbol{t} + \psi \cdot m) - e^{-\tilde{m} \cdot \epsilon} \mathcal{F}_\epsilon(\boldsymbol{t} + \psi \cdot \tilde{m}) + e^{-m \cdot \epsilon - \tilde{m} \cdot \epsilon} \mathcal{F}_\epsilon(\boldsymbol{t} + \psi \cdot m + \psi \cdot \tilde{m})
$$
$$
= \wp_d(\boldsymbol{t}, m, \tilde{m}) + O(\epsilon) = \text{regular in } \epsilon, \quad (6.2)
$$

where $m$ and $\tilde{m}$ correspond to fixed toric divisors $D$ and $\tilde{D}$ that have a compact intersection and $\wp$ is a polynomial.[6] Analogously, one can derive the quantum version of second-order shift equations, and discuss quantum corrections. We are not going to write explicit formulas, as they are straightforward by now. At this point, we do not understand the geometric meaning of these second-order shift equations. We provide a couple of examples below.

Let us comment on the analog of eq. (4.21) for the present case. If we fix two divisors $D_I$ and $D_J$ such that their intersection $D_I D_J$ is compact, then we have the expansion

$$
Z_q(\boldsymbol{T}) = \sum_{n,m=0}^\infty q_{D_I}^n q_{D_J}^m Z_q^{D_I D_J}(\boldsymbol{T} + \psi \cdot n + \psi \cdot m), \quad (6.3)
$$

where $q_{D_I}, q_{D_J}$ are equivariant parameters corresponding to divisors $D_I$ and $D_J$ and $Z_q^{D_I D_J}(\boldsymbol{T})$ is formally the partition function of the compact intersection of divisors $D_I D_J$ and thus it is a polynomial. However, in the above sum $Z_q^{D_I D_J}$ can be a different polynomial for different parts of sum, due to chamber issues related to the values of $\boldsymbol{T} + \psi \cdot n + \psi \cdot m$.

## 6.1 Resolved conifold

The simplest example $X$ with empty $H^2_{\text{comp}}(X)$ and non-empty $H^4_{\text{comp}}(X)$ is the resolved conifold. This Calabi-Yau three-fold is the resolution of the conifold singularity in $\mathbb{C}^4$, and it corresponds to the total space of $\mathcal{O}(-1) \oplus \mathcal{O}(-1) \to \mathbb{P}^1$. For this space $H^2_{\text{comp}}(X) = H_4(X)$ is empty, and $H^4_{\text{comp}}(X) = H_2(X)$ is one-dimensional.

This $X$ is the quotient of $\mathbb{C}^4$ by $U(1)$ with $Q = (1, 1, -1, -1)$. Momenta satisfy

$$
p_1 + p_2 - p_3 - p_4 = t. \quad (6.4)
$$

The intersection of divisors $X \cap \{p_3 = 0\}$ and $X \cap \{p_4 = 0\}$ is compact, and by Poincaré duality it generates $H^4_{\text{comp}}(X)$. The equivariant volume is

$$
\mathcal{F}_\epsilon(t) = \frac{e^{-\epsilon_1 t}}{(\epsilon_2 - \epsilon_1)(\epsilon_3 + \epsilon_1)(\epsilon_4 + \epsilon_1)} + \frac{\epsilon^{-\epsilon_2 t}}{(\epsilon_1 - \epsilon_2)(\epsilon_3 + \epsilon_2)(\epsilon_4 + \epsilon_2)}. \quad (6.5)
$$

Equation (6.2) becomes

$$
\mathcal{F}_\epsilon(t) - e^{-m\epsilon_3} \mathcal{F}_\epsilon(t + m) - e^{-\tilde{m}\epsilon_4} \mathcal{F}_\epsilon(t + \tilde{m}) + e^{-m\epsilon_3 - \tilde{m}\epsilon_4} \mathcal{F}_\epsilon(t + m + \tilde{m})
$$
$$
= t m \tilde{m} + \frac{1}{2} m^2 \tilde{m} + \frac{1}{2} m \tilde{m}^2 + O(\epsilon). \quad (6.6)
$$

---

[6]With slight abuse of notation, we use the same letters $\wp$ and $\mathcal{P}$ for the higher cohomologies.

There is no natural way to solve this equation, if we want to stay in the same chamber $t > 0$. The partition function

$$Z_q(T) = \sum_{s,n,k,l \geq 0,\, s+n-k-l=T} q_1^s q_2^n q_3^m q_4^l \tag{6.7}$$

can be rewritten as

$$Z_q(T) = \sum_{k=0}^{\infty}\sum_{l=0}^{\infty} q_3^k q_4^l \sum_{s=0}^{T+k+l} q_1^s q_2^{T+k+l-s} = \sum_{k=0}^{\infty}\sum_{l=0}^{\infty} q_3^k q_4^l Z_{q_1,q_2}^{\mathbb{P}^1}(T+k+l), \tag{6.8}$$

which illustrates the statement in eq. (6.3). One can derive the quantum analog of eq. (6.6)

$$Z_q(T) - q_3^M Z_q(T+M) - q_4^{\tilde{M}} Z_q(T+\tilde{M}) + q_3^M q_4^{\tilde{M}} Z_q(T+M+\tilde{M}) = \mathcal{P}(T,M,\tilde{M};q), \tag{6.9}$$

where $T, M, \tilde{M}$ are integers and $\mathcal{P}$ is a polynomial in $q$'s,

$$\mathcal{P}(T,M,\tilde{M};q) = \sum_{k=0}^{M-1}\sum_{l=0}^{\tilde{M}-1} q_3^k q_4^l Z_{q_1,q_2}^{\mathbb{P}^1}(T+k+l) \tag{6.10}$$

such that

$$\mathcal{P}(T,M,\tilde{M};1) = TM\tilde{M} + \frac{1}{2}M\tilde{M}^2 + \frac{1}{2}M^2\tilde{M}, \tag{6.11}$$

which agrees with the RHS of eq. (6.6). There are no quantum corrections.

## 6.2 Calabi-Yau four-fold

As another example $X$ with empty $H^2_{\text{comp}}(X)$ and non-empty $H^4_{\text{comp}}(X)$, consider the Calabi-Yau four-fold given by the quotient $\mathbb{C}^6 // U(1)^2$ with

$$Q = \begin{pmatrix} 1 & 1 & 0 & 0 & -1 & -1 \\ 0 & 0 & 1 & 1 & -1 & -1 \end{pmatrix}. \tag{6.12}$$

Momenta satisfy

$$\begin{aligned} p_1 + p_2 - p_5 - p_6 &= t_1, \\ p_3 + p_4 - p_5 - p_6 &= t_2. \end{aligned} \tag{6.13}$$

It has empty $H_6(X) = H^2_{\text{comp}}(X)$. However, $H_4(X) = H^4_{\text{comp}}(X)$ is one-dimensional, generated by the intersection of divisors $X \cap \{p_5 = 0\}$ and $X \cap \{p_6 = 0\}$. The equivariant volume is

$$\mathcal{F}_\epsilon(t) = \int \frac{d\phi_1 d\phi_2}{(2\pi i)^2} \frac{e^{\phi_1 t_1} e^{\phi_2 t_2}}{(\phi_1 + \epsilon_1)(\phi_1 + \epsilon_2)(\epsilon_5 - \phi_1 - \phi_2)(\epsilon_6 - \phi_1 - \phi_2)(\phi_2 + \epsilon_3)(\phi_2 + \epsilon_4)}. \tag{6.14}$$

By a direct calculation, we can derive the second-order shift equation

$$\begin{aligned} \mathcal{F}_\epsilon(t_1, t_2) - e^{-m\epsilon_5}\mathcal{F}_\epsilon(t_1 + m, t_2 + m) &- e^{-\tilde{m}\epsilon_6}\mathcal{F}_\epsilon(t_1 + \tilde{m}, t_2 + \tilde{m}) \\ + e^{-m\epsilon_5 - \tilde{m}\epsilon_6}\mathcal{F}_\epsilon(t_1 + m + \tilde{m}, t_2 + m + \tilde{m}) & \\ = m\tilde{m}t_1 t_2 + \frac{1}{2}(t_1 + t_2)\left(m\tilde{m}^2 + m^2\tilde{m}\right) &+ \frac{1}{2}m^2\tilde{m}^2 + \frac{1}{3}\left(m^3\tilde{m} + m\tilde{m}^3\right) + O(\epsilon). \end{aligned} \tag{6.15}$$

We are not sure how to analyze this equation. The partition function

$$Z_q(T_1, T_2) = \sum_{k=0}^{\infty}\sum_{l=0}^{\infty} q_5^k q_6^l Z_{q_1,q_2}^{\mathbb{P}^1}(T_1+k+l) Z_{q_3,q_4}^{\mathbb{P}^1}(T_2+k+l) \tag{6.16}$$

satisfies the quantum analog of eq. (6.15)

$$Z_q(\boldsymbol{T}) - q_5^M Z_q(T_1 + M, T_2 + M) - q_6^{\tilde{M}} Z_q(T_1 + \tilde{M}, T_2 + \tilde{M}) + q_5^M q_6^{\tilde{M}} Z_q(T_1 + M + \tilde{M}, T_2 + M + \tilde{M})$$

$$= \mathcal{P}(T_1, T_2, M, \tilde{M}; q) = \sum_{k=0}^{M-1} \sum_{l=0}^{\tilde{M}-1} q_5^k q_6^l \, Z_{q_1, q_2}^{\mathbb{P}^1}(T_1 + k + l) Z_{q_3, q_4}^{\mathbb{P}^1}(T_2 + k + l), \quad (6.17)$$

where $\mathcal{P}$ is a polynomial in $q$'s, and $M_1, M_2$ are integers. If we evaluate it at $q = 1$, we get

$$\mathcal{P}(T_1, T_2, M, \tilde{M}; 1) =$$
$$M\tilde{M} T_1 T_2 + \frac{1}{2}(T_1 + T_2)\left(M\tilde{M}^2 + M^2\tilde{M}\right) + \frac{1}{2}M^2\tilde{M}^2 + \frac{1}{3}\left(M^3\tilde{M} + M\tilde{M}^3\right) + \frac{2}{3}M\tilde{M}, \quad (6.18)$$

where the last term corresponds to a quantum correction.

# 7 Higher times

So far we concentrated on the equivariant volume $\mathcal{F}_\epsilon(\boldsymbol{t})$, as defined in section 2. We can also insert under the integral monomials $\mathrm{Mon}_k(\phi)$ of degree $k$ in $\phi$

$$\oint_C \prod_{a=1}^r \frac{d\phi_a}{2\pi\mathrm{i}} \frac{e^{t^a \phi_a} \mathrm{Mon}_k(\phi)}{\prod_{i=1}^N (\epsilon_i + Q_i^a \phi_a)} = \int_X e^{\omega + H}(G_{2k} + \cdots + G_0), \quad (7.1)$$

which can be interpreted as insertion of the equivariant class $(G_{2k} + \cdots + G_0)$ of degree $2k$ in the DH formula on $X$. These integrals are calculated in the same fashion as we discussed previously (i.e., keeping the same integration contour as before). To encode these classes, it is convenient to define a generating function for higher times

$$\mathcal{F}_\epsilon(\boldsymbol{t}; \boldsymbol{t}^{(2)}) = \oint_C \prod_{a=1}^r \frac{d\phi_a}{2\pi\mathrm{i}} \frac{\exp(t^a \phi_a + t^{ab} \phi_a \phi_b)}{\prod_{i=1}^N (\epsilon_i + Q_i^a \phi_a)}, \quad (7.2)$$

where we introduced higher times $\boldsymbol{t}^{(2)}$ (the generalization to other times $\boldsymbol{t}^{(n)}$ is straightforward). The generating function $\mathcal{F}_\epsilon(\boldsymbol{t}; \boldsymbol{t}^{(2)})$ satisfies many relations, e.g.

$$\left(\frac{\partial^2}{\partial t^a \partial t^b} - \frac{\partial}{\partial t^{ab}}\right) \mathcal{F}_\epsilon(\boldsymbol{t}; \boldsymbol{t}^{(2)}) = 0 \quad (7.3)$$

and at first it may appear to be a redundant object. However, as we explain in next section, in some special setting it can lead to interesting results.

The generating function with higher times can be used to write shift equations corresponding to the action of higher compact support cohomologies. Let us concentrate on $H^4_{\mathrm{comp}}(X)$, as the generalization to higher cohomologies (higher times) is straightforward. For $X$ with non-empty $H^4_{\mathrm{comp}}(X)$, we follow a similar route to the previous sections, and write

$$\mathcal{F}_\epsilon(\boldsymbol{t}; \boldsymbol{t}^{(2)}) - e^{m^{IJ}\epsilon_I\epsilon_J} \mathcal{F}_\epsilon(t^a + \epsilon_J Q_I^a m^{IJ} + \epsilon_I Q_J^a m^{IJ}; t^{ab} + Q_I^a Q_J^b m^{IJ})$$
$$= \text{polynomial in } (\boldsymbol{t}, \boldsymbol{t}^{(2)}; m^{(2)}) + O(\epsilon), \quad (7.4)$$

where we are allowed to sum over pairs $(I, J)$ such that divisors $(D_I, D_J)$ have a compact intersection. This is an important difference compared to eq. (6.1). It is useful to extend the definition of $\psi$-map to higher times:

$$\psi : t^{ab} \mapsto t^{ab} + \sum_{I,J} Q_I^a Q_J^b m^{IJ} \quad (7.5)$$

so that we can rewrite eq. (7.4) as

$$\mathcal{F}_\epsilon(t; t^{(2)}) - e^{\epsilon \cdot m^{(2)} \cdot \epsilon} \mathcal{F}_\epsilon(t + \psi \cdot (\epsilon \cdot m^{(2)}); t^{(2)} + \psi \cdot m^{(2)}) = \text{polynomial in } (t, t^{(2)}; m^{(2)}) + O(\epsilon).$$
(7.6)

To prove this equation, it is enough to analyze the pole structure of the LHS and observe that the problem becomes compact. The main difference with our previous discussion of $H^4_{\text{comp}}$ in section 6 is that the equation with higher times is a first-order difference equation while the shift equations in section 6 are second-order difference equations.

# 8 Calabi-Yau five-folds and factorization

The construction in this section is motivated by the study of the classical action for higher-rank Donaldson-Thomas theory [19]. Consider the Calabi-Yau five-fold realized as the product $CY_5 = CY_3 \times A_{n-1}$ of four-dimensional $A_{n-1}$ space and a toric Calabi-Yau three-fold with non-empty $H^2_{\text{comp}}(CY_3) \approx H_4(CY_3)$. For $CY_5$, $H_8(CY_5) = H^2_{\text{comp}}(CY_5)$ is empty, but $H_6(CY_5) = H^4_{\text{comp}}(CY_5)$ is non-empty.

Our $CY_5$ is toric and it comes from the quotient

$$CY_5 = CY_3 \times A_{n-1} = \mathbb{C}^{r+3}//U(1)^r \times \mathbb{C}^{n+1}//U(1)^{n-1} = \mathbb{C}^{n+r+4}//U(1)^{n+r-1},$$
(8.1)

where we use the same notations as before. To distinguish the two factors, we put tildes over all objects for $A_{n-1}$ space. The $CY_3$ has compact divisors $\{p^I = 0\}$ for $I = 1, \ldots, \dim H^2_{\text{comp}}(CY_3)$, while $A_{n-1}$ has compact divisors $\{\tilde{p}^{\tilde{I}} = 0\}$ for $\tilde{I} = 1, \ldots, n-1$. Therefore, on $CY_5$ the intersections of divisors $\{p^I = 0\} \cap \{\tilde{p}^{\tilde{I}} = 0\}$ are compact and generate $H^4_{\text{comp}}(CY_5)$.

The equivariant volume of $CY_5$ is

$$\mathcal{F}^{CY_5}_{\epsilon, \tilde{\epsilon}}(t, \tilde{t}) = \oint_C \prod_{a=1}^r \frac{d\phi_a}{2\pi i} \prod_{\tilde{a}=1}^{n-1} \frac{d\tilde{\phi}_{\tilde{a}}}{2\pi i} \frac{e^{t^a \phi_a + \tilde{t}^{\tilde{a}} \tilde{\phi}_{\tilde{a}}}}{\prod_{i=1}^{r+3}\left(\epsilon_i + Q^a_i \phi_a\right) \prod_{\tilde{i}=1}^{n+1}\left(\tilde{\epsilon}_{\tilde{i}} + \tilde{Q}^{\tilde{a}}_{\tilde{i}} \tilde{\phi}_{\tilde{a}}\right)}.$$
(8.2)

Following ideas from section 6, we could write a second-order difference equation for $\mathcal{F}^{CY_5}_{\epsilon, \tilde{\epsilon}}(t, \tilde{t})$ corresponding to the action of $H^4_{\text{comp}}(CY_5)$. Instead, we define the equivariant volume with higher times as outlined in previous section. We only turn on the off-diagonal higher times, with legs along compact divisors of each factor. The explicit form is

$$\mathcal{F}^{CY_5}_{\epsilon, \tilde{\epsilon}}(t, \tilde{t}; \psi \cdot m^{(2)}) = \oint_C \prod_{a=1}^r \frac{d\phi_a}{2\pi i} \prod_{\tilde{a}=1}^{n-1} \frac{d\tilde{\phi}_{\tilde{a}}}{2\pi i} \frac{e^{t^a \phi_a + \tilde{t}^{\tilde{a}} \tilde{\phi}_{\tilde{a}} + m^{I\tilde{J}} Q^a_I \tilde{Q}^{\tilde{b}}_{\tilde{J}} \phi_a \tilde{\phi}_{\tilde{b}}}}{\prod_{i=1}^{r+3}\left(\epsilon_i + Q^a_i \phi_a\right) \prod_{\tilde{i}=1}^{n+1}\left(\tilde{\epsilon}_{\tilde{i}} + \tilde{Q}^{\tilde{a}}_{\tilde{i}} \tilde{\phi}_{\tilde{a}}\right)},$$
(8.3)

where we introduced higher times $m^{(2)}$. This object is well-defined if we use the standard JK prescription for each factor. Using eq. (7.4), we can write a shift equation, which is controlled by $H^4_{\text{comp}}(CY_5)$:

$$e^{\epsilon_I \tilde{\epsilon}_{\tilde{J}} m^{I\tilde{J}}} \mathcal{F}^{CY_5}_{\epsilon, \tilde{\epsilon}}\left(t^a + m^{I\tilde{J}} Q^a_I \tilde{\epsilon}_{\tilde{J}}, \tilde{t}^{\tilde{b}} + m^{I\tilde{J}} \tilde{Q}^{\tilde{b}}_{\tilde{J}} \epsilon_I, m^{I\tilde{J}} Q^a_I \tilde{Q}^{\tilde{b}}_{\tilde{J}}\right) - \mathcal{F}^{CY_5}_{\epsilon, \tilde{\epsilon}}\left(t^a, \tilde{t}^{\tilde{b}}, 0\right)$$
$$= \text{polynomial in } (t, \tilde{t}; m^{(2)}) + O(\epsilon, \tilde{\epsilon}), \quad (8.4)$$

which can be derived by analyzing the pole structure of LHS, as in our previous discussion.

To simplify our discussion, we set $\epsilon_I = 0$ and $\tilde{\epsilon}_{\tilde{j}} = 0$ (for all compact divisors of $CY_3$ and $A_{n-1}$).[7] If we also set $\tilde{t} = 0$, we are left with

$$\mathcal{F}_{\epsilon,\tilde{\epsilon}}^{CY_5}(\boldsymbol{t}, 0; \psi \cdot m^{(2)}) = \oint_C \prod_{a=1}^r \frac{d\phi_a}{2\pi i} \prod_{\tilde{a}=1}^{n-1} \frac{d\tilde{\phi}_{\tilde{a}}}{2\pi i} \frac{\exp\left(t^a \phi_a + Q_I^a (m^{I\tilde{J}} \tilde{Q}_{\tilde{j}}^{\tilde{b}} \tilde{\phi}_{\tilde{b}}) \phi_a\right)}{\prod_{i=1}^{r+3} \left(\epsilon_i + Q_i^a \phi_a\right) \prod_{i=1}^{n+1} \left(\tilde{\epsilon}_{\tilde{i}} + \tilde{Q}_{\tilde{i}}^{\tilde{a}} \tilde{\phi}_{\tilde{a}}\right)}, \tag{8.5}$$

where higher times $m^{I\tilde{J}} \tilde{Q}_{\tilde{j}}^{\tilde{b}}$ can be thought of as Kähler parameters on $A_{n-1}$ with an extra label corresponding to compact divisors of $CY_3$. The explicit evaluation of eq. (8.5) gives

$$\mathcal{F}_{\epsilon,\tilde{\epsilon}}^{CY_5}(\boldsymbol{t}, 0; \psi \cdot m^{(2)}) = \sum_{p=1}^n \frac{1}{\varepsilon_4^{(p)} \varepsilon_5^{(p)}} \mathcal{F}_\epsilon^{CY_3}(\boldsymbol{t} - \psi \cdot H^p) \tag{8.6}$$

as a sum over fixed points of $A_{n-1}$ space, where $(\varepsilon_4^{(p)}, \varepsilon_5^{(p)})$ are local equivariant parameters at fixed point $p$ (see Appendix A.2 in ref. [19]) and $H^p$ is the value of a family of $A_{n-1}$ Hamiltonians (parametrized by compact divisors on $CY_3$) at point $p$. These are defined in eq. (B.8), and their relation to $m^{(2)}$ is given by the change of basis

$$m^{I\tilde{J}} \tilde{Q}_{\tilde{j}}^{\tilde{a}} = \alpha^{I(n-\tilde{a}+1)} - \alpha^{I(n-\tilde{a})}. \tag{8.7}$$

To make this map one-to-one, we impose the condition $\sum_{p=1}^n \alpha^{Ip} = 0$ for every $I$. Switching from $m^{(2)}$'s to $\alpha$'s makes formulas simpler (this is the standard trick relating $su(n)$ Cartan matrix to the diagonal $u(n)$ Cartan matrix).

Rewriting eq. (8.4) for this case, we get

$$\mathcal{F}_{\epsilon,\tilde{\epsilon}}^{CY_5}(\boldsymbol{t}, 0; \psi \cdot m^{(2)}) - \mathcal{F}_{\epsilon,\tilde{\epsilon}}^{CY_5}(\boldsymbol{t}, 0; 0) = \mathcal{P}^{CY_5}(\boldsymbol{t}; m^{(2)}) + O(\epsilon, \tilde{\epsilon}) \tag{8.8}$$

and our goal is to calculate the polynomial $\mathcal{P}^{CY_5}$ in terms of $CY_3$ data.

## 8.1 $CY_3$ data

We recall some facts about toric Calabi-Yau three-folds and set the notations for further discussions. Let $CY_3$ be a toric Calabi-Yau three-fold ($d = 3$) with non-empty $H_{\text{comp}}^2(CY_3)$. Expanding in powers of $\hbar$ the difference of volumes at $\boldsymbol{T}$ and $\boldsymbol{T} + \psi \cdot M$, using eqs. (3.8) and (3.10), we get

$$\mathcal{P}^{CY_3}(\boldsymbol{T}, M; q) = \frac{1}{\hbar^d} \oint_C \prod_{a=1}^r \frac{d\phi_a}{2\pi i} \frac{e^{\phi \cdot t} - e^{\phi \cdot (t+\psi \cdot m)} \prod_I q_I^{M^I}}{\prod_i x_i} \left(1 + \frac{\hbar^2}{12} c_2 + O(\hbar^4)\right). \tag{8.9}$$

The Calabi-Yau condition implies $c_1 = \sum_i x_i = \sum_i \epsilon^i$, so $c_1$ and $c_1^2$ are $O(\epsilon)$ and $O(\epsilon^2)$, respectively; being independent of $\phi$, they factor out of the integral, and contribute higher orders in $\hbar$, since eq. (4.7) implies (here we set $\epsilon$'s to zero along the compact divisors)

$$\mathcal{F}_\epsilon^{CY_3}(\boldsymbol{t}) - \mathcal{F}_\epsilon^{CY_3}(\boldsymbol{t} + \psi \cdot m) = \text{a cubic polynomial } \mathcal{P}_3^{CY_3}(\boldsymbol{t}, m) + O(\hbar^4). \tag{8.10}$$

That's because $O(\epsilon)$ terms come with higher-degree polynomials in $(\boldsymbol{t}, m)$. We denote by $\mathcal{P}_{p,q}^{CY_3}$ the part of $\mathcal{P}_3^{CY_3}$ of degree $p$ in $\boldsymbol{t}$ and $q$ in $m$, with $p + q = 3$

$$\mathcal{P}_3^{CY_3}(\boldsymbol{t}, m) = \mathcal{P}_{0,3}^{CY_3}(m) + \mathcal{P}_{1,2}^{CY_3}(\boldsymbol{t}, m) + \mathcal{P}_{2,3}^{CY_3}(\boldsymbol{t}, m)$$
$$= A_{IJK} m^I m^J m^K + B_{aIJ} t^a m^I m^J + C_{abI} t^a t^b m^I \tag{8.11}$$

---

[7]This is without loss of generality, as they can be turned back on using eq. (2.20). We also keep denoting the subscript indices in $\mathcal{F}$ in the same way, with the understanding that some of the $\epsilon$'s may be zero.

and clearly $\mathcal{P}_{3,0}^{CY_3} = 0$. If we define

$$\mathcal{C}_\epsilon^{CY_3}(t) = \frac{1}{12} \oint_C \prod_{a=1}^r \frac{d\phi_a}{2\pi i} \frac{e^{\phi \cdot t}}{\prod_i x_i} c_2 \tag{8.12}$$

then this satisfies

$$\mathcal{C}_\epsilon^{CY_3}(t) - \mathcal{C}_\epsilon^{CY_3}(t + \psi \cdot m) = \text{a linear polynomial } \mathcal{P}_1^{CY_3}(m) + O(\hbar^2) \tag{8.13}$$

and $\mathcal{P}_1$ is a function of $m$ only. So we get

$$\mathcal{P}^{CY_3}(T, M; q) = \frac{\mathcal{P}_3^{CY_3}(t, m)}{\hbar^3} + \frac{\mathcal{P}_1^{CY_3}(m)}{\hbar} + O(\hbar). \tag{8.14}$$

Taking the limit $\hbar \to 0$, with fixed $T = t/\hbar$ and $M = m/\hbar$, we get eq. (4.20),

$$\mathcal{P}^{CY_3}(T, M; 1) = \mathcal{P}_3^{CY_3}(T, M) + \mathcal{P}_1^{CY_3}(M). \tag{8.15}$$

As we explained in section 4.2, if we evaluate $\mathcal{P}_3^{CY_3}(t, m)$ at some specific $m$'s, we obtain the analog of triple intersection on $CY_3$, which is a genus-zero contribution from constant maps in Gromov-Witten theory. Evaluating $\mathcal{P}_1^{CY_3}(m)$ at the same $m$'s gives the genus-one contribution from constant maps in Gromov-Witten theory. As we explained, these objects are not unique but instead depend on some choices.

## 8.2 Shift equations for $CY_5$

We calculate $\mathcal{P}^{CY_5}(t; m^{(2)})$ in eq. (8.8) in terms of $CY_3$ data. Combining eqs. (8.6) and (8.10), we obtain

$$\mathcal{F}_{\epsilon,\tilde{\epsilon}}^{CY_5}(t, 0; \psi \cdot m^{(2)}) = \sum_{p=1}^n \frac{1}{\varepsilon_4^{(p)} \varepsilon_5^{(p)}} \mathcal{F}_\epsilon^{CY_3}(t) - \sum_{p=1}^n \frac{1}{\varepsilon_4^{(p)} \varepsilon_5^{(p)}} \mathcal{P}_3^{CY_3}(t, -H^p) + O(\epsilon). \tag{8.16}$$

Observing that the first term in RHS is $\mathcal{F}_{\epsilon,\tilde{\epsilon}}^{CY_5}(t, 0; 0)$, we get

$$\mathcal{P}^{CY_5}(t; m^{(2)}) = - \sum_{p=1}^n \frac{1}{\varepsilon_4^{(p)} \varepsilon_5^{(p)}} \mathcal{P}_3^{CY_3}(t, -H^p). \tag{8.17}$$

We can write this more explicitly

$$\mathcal{P}^{CY_5}(t; m^{(2)}) = \sum_{p=1}^n \frac{1}{\varepsilon_4^{(p)} \varepsilon_5^{(p)}} \left( A_{IJK} H^{Ip} H^{Jp} H^{Kp} - B_{aIJ} t^a H^{Ip} H^{Jp} + C_{abI} t^a t^b H^{Ip} \right). \tag{8.18}$$

Next we use some combinatorial properties of $A_{n-1}$ space (see appendix B here and appendix C in ref. [19]), to obtain

$$\mathcal{F}_{\epsilon,\tilde{\epsilon}}^{CY_5}(t, 0; \psi \cdot m^{(2)}) = \mathcal{F}_{\epsilon,\tilde{\epsilon}}^{CY_5}(t, 0; 0) + \sum_{p=1}^n \mathcal{P}_{1,2}^{CY_3}(t, \alpha^p) + g \sum_{p=1}^n \mathcal{P}_{0,3}^{CY_3}(\alpha^p)$$
$$+ \frac{\varepsilon}{2} \sum_{q<p} \mathcal{P}_{0,3}^{CY_3}(\alpha^q - \alpha^p) + O(\epsilon), \tag{8.19}$$

where $g$ and $\varepsilon$ are combinations of equivariant parameters on $A_{n-1}$ (see appendix B).

## 8.3 Factorization in higher rank DT theory

In this subsection, the discussion is within the context of the work [19], where we studied rank $n$ K-theoretic Donaldson-Thomas (DT) theory on a toric threefold $CY_3$ and we conjectured certain factorization properties for the classical action of this theory. As a corollary of eq. (8.19), we show this factorization property for any toric $CY_3$ with non-empty $H^2_{\text{comp}}(CY_3)$. We apply our previous discussion to prove equality 5.29 in ref. [19] (see also appendix C there). We define the classical action for $U(n)$ DT theory[8] on $CY_3$ as

$$u_n(\alpha; g, t) = \frac{n \mathcal{F}_\epsilon^{CY_3}(t)}{(n\epsilon/2)^2 - g^2} + \sum_{p=1}^n \mathcal{P}_{1,2}^{CY_3}(t, \alpha^p) + g \sum_{p=1}^n \mathcal{P}_{0,3}^{CY_3}(\alpha^p) - \frac{n}{2} \mathcal{C}_\epsilon^{CY_3}(t), \qquad (8.20)$$

where the last term is defined in eq. (8.12). We drop the dependence on partitions $K$, $\lambda$, which is irrelevant to our present discussion. In the context of DT theory, $g$ and $\varepsilon$ can be regarded as couplings, but they are related to the CY3 $\epsilon$'s in a natural way: if one takes into account the Calabi-Yau-5 picture, then $g$ and $\varepsilon$ are specific combinations of equivariant parameters of the $A_{n-1}$ part of $CY_5$. The equality we want to prove is

$$u_n(\alpha; g, t) + \frac{\varepsilon}{2} \sum_{q<p} \mathcal{P}^{CY_3}(0, \alpha^q - \alpha^p; 1) = \sum_{p=1}^n u_1(0; g_p, t_p), \qquad (8.21)$$

where $\mathcal{P}^{CY_3}(0, \alpha_{qp}; 1)$ is defined in eq. (8.15), and it corresponds to perturbative contributions in ref. [19], where we denoted it by $|\mathcal{P}_{qp}|$. In eq. (8.21) we set (no summation over $p$)

$$t^p = t + \psi \cdot (g_p \alpha^p + \frac{\varepsilon}{2} \sigma^p), \quad g_p = g + \frac{\varepsilon}{2}(n + 1 - 2p), \qquad (8.22)$$

where we use notations from appendix B and $\psi$ corresponds to the $\psi$-map for $CY_3$. Equation (8.21) describes factorization of $U(n)$ gauge theory on $CY_3$ into copies of $U(1)$ theory.

To prove eq. (8.21), we combine eqs. (8.13) and (8.19)

$$\sum_{p=1}^n u_1(0; g_p, t^p) - \frac{n \mathcal{F}_\epsilon^{CY_3}(t)}{(n\epsilon/2)^2 - g^2} + \frac{n}{2} \mathcal{C}_\epsilon^{CY_3}(t) =$$
$$\frac{\varepsilon}{2} \sum_{q<p} \mathcal{P}_{0,3}^{CY_3}(\alpha^q - \alpha^p) + g \sum_{p=1}^n \mathcal{P}_{0,3}^{CY_3}(\alpha^p) + \sum_{p=1}^n \mathcal{P}_{1,2}^{CY_3}(t, \alpha^p) + \sum_{p=1}^n \mathcal{P}_1^{CY_3}(g_p \alpha^p + \frac{\varepsilon}{2} \sigma_p) + O(\epsilon).$$
$$(8.23)$$

Using eq. (8.14), namely

$$\sum_{q<p} \mathcal{P}^{CY_3}(0, \alpha^q - \alpha^p; 1) = \sum_{q<p} \mathcal{P}_{0,3}^{CY_3}(t, \alpha^q - \alpha^p) + \sum_{q<p} \mathcal{P}_1^{CY_3}(\alpha^q - \alpha^p) \qquad (8.24)$$

we obtain eq. (8.21) up to order $O(\epsilon)$. If we choose the truncated $\hat{\mathcal{F}}_\epsilon^{CY_3}(t)$, which satisfies the shift equation exactly, then

$$\hat{\mathcal{F}}_\epsilon^{CY_3}(t) - \hat{\mathcal{F}}_\epsilon^{CY_3}(t + \psi \cdot m) = \mathcal{P}_3^{CY_3}(t, m) \qquad (8.25)$$

and using $\hat{\mathcal{F}}_\epsilon^{CY_3}(t)$ in eq. (8.20) as classical action we get eq. (8.21) exactly. As described in section 4.2, upon certain choices one can solve the shift equation: for example, we can choose as $\hat{\mathcal{F}}_\epsilon^{CY_3}(t)$ the polynomial $\wp_d(t^\alpha, (Q_{-1})_J^I t^J)$ (see section 4.2 and the examples in sections 5.3

---

[8]We previously set $\sum \alpha = 0$, so we are actually dealing with $SU(n)$ theory. Equation (8.21) below is also valid modulo $\sum \alpha$ terms.

and 5.4). Alternatively, we can keep the singular terms in $\epsilon$'s together with $\wp_d(t_\alpha, (Q^{-1})^J_I t_J)$ as part of $\hat{\mathcal{F}}^{CY_3}_\epsilon(t)$, and this still satisfies the shift equation exactly.

In ref. [19], the numbers $\alpha^{Ip}$ were assumed to be integers. In the present context the integrality of $\alpha^{Ip}$ does not play any role, although we used the semi-classical expansion for $CY_3$. Understanding this better would require embedding the Calabi-Yau fivefold picture into a quantum mechanical framework. At the moment, we do not know how to do this consistently in the presence of higher times.

## 8.4 $M$-theory interpretation

The classical action $u_n(\alpha; g, t)$ is related to the equivariant volume with higher times $\mathcal{F}^{CY_5}_{\epsilon, \tilde{\epsilon}}(t, 0; \psi \cdot m^{(2)})$, defined in eq. (8.3) for $CY_5 = A_{n-1} \times CY_3$ (many arguments here can be extended to generic toric $CY_5$). They differ by the semi-classical part $\mathcal{C}^{CY_3}_\epsilon(t)$ and the perturbative part of DT theory. We discuss a possible interpretation of $\mathcal{F}^{CY_5}_{\epsilon, \tilde{\epsilon}}$ as equivariant Chern-Simons term in $M$-theory, as anticipated in eq. (1.10). On $CY_5$, the object

$$\mathcal{F}^{CY_5}_{\epsilon, \tilde{\epsilon}}(t, \tilde{t}; \psi.m^{(2)}) = \int_{CY_5} e^{G_4 + G_2 + G_0} \tag{8.26}$$

corresponds to the exponent of some equivariant form

$$(d + \iota_v)(G_4 + G_2 + G_0) = 0. \tag{8.27}$$

The choice of times $(t, \tilde{t}, m^{(2)})$ is related to the choice of equivariant forms of degree 2 and 4. With polar coordinates $(r, \phi)$ on the disk $D^2$, there is a natural action of $\partial_\phi + v$, with $v$ being the toric action on $CY_5$, and the equivariant form eq. (8.27) admits a lift to $D^2 \times CY_5$

$$(d + \iota_{\partial_\phi} + \iota_v)\big(G_4 + G_2 - d(G_0 r^2 d\phi) + G_0(1 - r^2)\big) = 0. \tag{8.28}$$

We can deform this lift, while preserving its class in equivariant cohomology, to some abstract equivariantly-closed form $G^{(4)} + G^{(2)} + G^{(0)}$ on $D^2 \times CY_5$, whose exponent provides the natural equivariant extension of the CS term. As the only fixed point on the disk is its center, by localization we have

$$\mathcal{F}^{CY_5}_{\epsilon, \tilde{\epsilon}}(t, \tilde{t}; \psi.m^{(2)}) = \int_{D^2 \times CY_5} \exp(G^{(4)} + G^{(2)} + G^{(0)}), \tag{8.29}$$

which makes eq. (1.10) into a fully equivariant object.

If we restrict eq. (8.28) from the disk to its boundary, then on $S^1 \times CY_5$ the zero-form part vanishes and the conditions for four-form and two-form parts coincide with the conditions on 11d supersymmetric solutions in ref. [39]. The two-form is constructed there as Killing spinor bilinear. This simple observation suggests a deeper relation between equivariance and 11d supersymmetric backgrounds, which deserves further study.

# 9 Conclusions

This work was provoked by ref. [19], where we tried to calculate triple intersection numbers for non-compact toric Calabi-Yau manifolds using DH formula. We were unable to reproduce the known results from the localization calculation. In this work we address this puzzle and suggest a framework to extract the intersections numbers using equivariant DH formula for non-compact toric Kähler manifolds. Unlike the compact case, for non-compact manifolds we cannot turn

off the equivariant parameters and extract easily geometrical data from the answer. Our main insight is that the full equivariant symplectic volume satisfies a difference equation (we refer to it as shift equation), which is controlled by the action of compact support cohomology on de Rham cohomology. Upon certain non-canonical choices we can solve this equation and extract the intersection polynomial. We also consider the quantum mechanical analog of shift equations and go through a number of explicit examples. As a byproduct, we prove a result about factorization of classical actions in the context of non-abelian Donaldson-Thomas theory on Calabi-Yau threefolds (conjectured in ref. [19]).

## 9.1 Future directions

Physically, it would be important to investigate further the 5d theory deformed by Calabi-Yau internal isometries, and the role of the equivariant volume (or some related quantity). Since extracting the intersection polynomial requires some non-canonical choices, it is suggestive that in the context of string theory the full equivariant symplectic volume of a toric Calabi-Yau threefold should be taken seriously and equivariant parameters should have a clear physical interpretation.

Mathematically, it would be interesting to extend our results to the quantum cohomology ring, and the study of Gromov-Witten invariants in the equivariant setup, without making ad hoc choices for the equivariant parameters.

## Acknowledgements

NN thanks K. Becker, M. Dedushenko, J. Kaidi and A. Losev for discussions. NP thanks L. Cassia and Y. Tachikawa for discussions. We thank M. Vergne for carefully reading the manuscript and providing an alternative proof of some of the results. NN is grateful to the IHES for hospitality in July 2021, when part of this work was done. The work of NP and MZ is supported by the grant "Geometry and Physics" from the Knut and Alice Wallenberg foundation. Opinions and conclusions expressed here are those of the authors and do not necessarily reflect the views of funding agencies.

## A Norms and quantum Kähler potential

We collect some facts about norms of states on Kähler quotients. On $\mathbb{C}^N$ define the norm

$$||\Psi||^2 = \int_{\mathbb{C}^N} \prod_{i=1}^{N} d^2 z^i \; |\Psi(z)|^2 e^{-\frac{1}{\hbar}\sum_{i=1}^{N}|z^i|^2}\,, \tag{A.1}$$

where the state $\Psi$ is a holomorphic function of $z$, or a monomial if we want it to be diagonal under all $U(1)$ actions. If we choose a $\Psi(z)$ that satisfies Gauss law, see eq. (3.1), then it descends to the quotient and it is interpreted as a section of the appropriate bundle. Let us see what happens to eq. (A.1) in this case. Under the integral in eq. (A.1), we can insert

$$\int \prod_{a=1}^{r} dV_a \; \det\|\sum_i Q_i^a Q_i^{a'}(e^{-Q_i V} p^i)\|_{aa'} \prod_b \delta(\mu^b(e^{-QV} \boldsymbol{p}) - t^b) = 1\,, \tag{A.2}$$

where $\mu^a(p)$ is defined in eq. (2.4) and we use the shorthand notation

$$e^{Q_i V} p^i \equiv \prod_{a=1}^{r} e^{Q_i^a V_a} p^i \tag{A.3}$$

suppressing the index $a$ when possible. We also introduced the real auxiliary variables $V_a$. Then we can do the change of variables $e^{-QV}\mathbf{p} \to \mathbf{p}$ under the integral. Assuming that $\Psi(\mathbf{z})$ satisfies eq. (3.1), we obtain

$$||\Psi||^2 = \int_{\mathbb{C}^N} \prod_{i=1}^{N} d^2 z^i \, |\Psi(\mathbf{z})|^2 e^{-K_q(\mathbf{p})} \det \| \sum_i Q_i^a Q_i^{a'} p^i \|_{aa'} \prod_b \delta(\mu^b(\mathbf{p}) - t^b), \qquad (A.4)$$

where $K_q$ is defined as

$$e^{-K_q(\mathbf{p})} := \int \prod_a dV_a \exp\left[ -\frac{1}{\hbar} \sum_i e^{Q_i V} p^i + T^a V_a + \sum_i Q_i^a V_a \right]. \qquad (A.5)$$

The last term in the exponent comes from the change in the measure and it is zero for Calabi-Yau quotients. By construction, eq. (A.4) gives a well-defined integral on the quotient with the appropriate Kähler form. To understand global issues, one can check that under $(\mathbb{C}^\times)$-action $z^i \mapsto (\lambda \cdot \mathbf{z})^i := z^i \prod_a \lambda_a^{Q_i^a}$ complemented by

$$V_a \to V_a - (\log \lambda_a + \log \bar{\lambda}_a) \qquad (A.6)$$

the potential $K_q(p)$ transforms as

$$K_q(\mathbf{p}) \to K_q(\mathbf{p}) + (T^a + \sum_i Q_i^a)(\log \lambda_a + \log \bar{\lambda}_a). \qquad (A.7)$$

This Kähler potential only agrees with the standard Kähler potential on the Kähler quotient [40] at leading order in the semi-classical expansion. The proper geometric and physical meaning of this quantum Kähler potential is not clear to us at the moment. In this appendix we just wanted to demonstrate that one can introduce an appropriate finite norm for the states that are diagonal under $U(1)$ action on non-compact toric spaces.

# B   Conventions for $A_{n-1}$ space

This appendix recalls some definitions for $A_{n-1}$ space, for a more detailed treatment the reader may consult Appendix A in ref. [19]. The space $A_{n-1}$ is $\mathbb{C}^{n+1}//U(1)^{n-1}$. On $\mathbb{C}^{n+1}$ we set to zero all $\tilde{\epsilon}$'s corresponding to compact divisors and keep only

$$\varepsilon_4 = \tilde{\epsilon}_1, \quad \varepsilon_5 = \tilde{\epsilon}_{n+1}. \qquad (B.1)$$

For $p = 1, \ldots, n$, we define equivariant parameters at fixed point $p$

$$\varepsilon_4^{(p)} = (n-p+1)\varepsilon_4 + (1-p)\varepsilon_5, \quad \varepsilon_5^{(p)} = (p-n)\varepsilon_4 + p\varepsilon_5. \qquad (B.2)$$

By DH theorem, the equivariant volume is

$$\text{vol}(A_{n-1}) = \sum_{p=1}^{n} \frac{e^{H^p}}{\varepsilon_4^{(p)} \varepsilon_5^{(p)}}, \qquad (B.3)$$

where $H^p$ is the value of Hamiltonian at fixed point $p$. One can compute

$$H^p = \varepsilon_4 \sum_{k=1}^{n-p} j\tilde{t}^k + \varepsilon_5 \sum_{k=1}^{p} (k-1)\tilde{t}^{n-k+1}, \qquad (B.4)$$

where the $(n-1)$ parameters $\tilde{t}$ correspond to the values of eq. (2.4), as described in Appendix A of ref. [19]. Introduce $\alpha^p$ such that

$$\tilde{t}^{n-p} = \alpha^{p+1} - \alpha^p\,. \tag{B.5}$$

This map is not invertible unless we add an extra condition. It is natural to require

$$\sum_{p=1}^{n} \alpha^p = 0\,. \tag{B.6}$$

One can check that eqs. (B.5) and (B.6) provide an invertible map between $\tilde{t}$ and $\alpha$'s. Using eq. (B.3) with the values of $H^p$ in eq. (B.4) expressed in terms of $\alpha$'s, we get

$$\mathrm{vol}(A_{n-1}) = \frac{1}{n\varepsilon_4\varepsilon_5} - \frac{1}{2}\sum_{p=1}^{n}(\alpha^p)^2 + \frac{\varepsilon_4 + \varepsilon_5}{12}\sum_{p<q}(\alpha^p - \alpha^q)^3 + \frac{n(\varepsilon_4 - \varepsilon_5)}{12}\sum_{p=1}^{n}(\alpha^p)^3 + O(\epsilon^2)\,. \tag{B.7}$$

Let us write $H^p$ in terms of $\alpha$'s

$$H^p = -\frac{\varepsilon}{2}\sigma^p(\alpha) - g_p \alpha^p\,, \tag{B.8}$$

where we used eq. (B.6) together with the map

$$\sigma^p(\alpha) := \sum_{s=1}^{p-1} \alpha^s - \sum_{s=p+1}^{n} \alpha^s \tag{B.9}$$

and we defined (from the point of view of this paper)

$$\varepsilon := \varepsilon_4 + \varepsilon_5 = \varepsilon_4^{(p)} + \varepsilon_5^{(p)}\,, \quad \varepsilon_4 - \varepsilon_5 =: \frac{2}{n}g \tag{B.10}$$

so that

$$g_p := g + \frac{\varepsilon}{2}(n+1-2p) = \frac{\varepsilon_4^{(p)} - \varepsilon_5^{(p)}}{2}\,. \tag{B.11}$$

## C  Appendix by Michèle Vergne

### C.1  Shifts in equivariant integration

N. Nekrasov, N. Piazzalunga and M. Zabzine (NPZ) discovered a shift equation for equivariant volumes of a family of Hamiltonian manifolds $X_{\mathbf{t}}$. The motivating example is: $X = \mathbb{C}^N$ with standard action of $T_N = U(1)^N$ and $X_{\mathbf{t}}$ the family of non compact toric manifolds arising by reduction with respect to an action of a subtorus $T$. Their proof uses an integral representation of the equivariant symplectic volume via Jeffrey-Kirwan residues. We outline a proof of this shift equation in a more general context, using equivariant cohomology arguments. We give a simple example in the last section, making explicit the cohomological arguments used.

We slightly changed notations, and stick to the notations of [41] (Chapter 7) for equivariant cohomology, so let us describe our setting.

Let $T_N = U(1)^N$ be the standard $N$ dimensional torus, with Lie algebra $\mathrm{Lie}(T_N)$. We denote by $\epsilon$ an element of $\mathrm{Lie}(T_N)$. We write $\epsilon = \sum_{i=1}^{N} \epsilon_i p^i$ where $\epsilon_i \in \mathbb{R}$ are reals. Let $X$ be a $T_N$-Hamiltonian manifold, possibly non compact, with symplectic form $\omega$ and moment map $\tilde{\mu} : X \to \mathrm{Lie}(T_N)^*$. The equivariant symplectic form is $\omega(\epsilon) = \omega + \langle \tilde{\mu}, \epsilon \rangle$ and satisfies $D_\epsilon \omega(\epsilon) = 0$

where $D_\epsilon = d - \iota(\epsilon_X)$. Here $\iota(\epsilon_X)$ is the contraction by the vector field $\epsilon_X$ associated to the infinitesimal action of $\epsilon$ on $X$.

Fix $T = U(1)^r$, and consider an injective homomorphism $T \to T_N$. Denote by $\Psi : \mathrm{Lie}(T_N)^* \to \mathrm{Lie}(T)^*$ the corresponding surjection. So the moment map $\mu : X \to \mathrm{Lie}(T)^*$ for the $T$-action is $\Psi\tilde{\mu}$. If $\mathbf{t}$ is a point in $\mathrm{Lie}(T)^*$, then $\mu^{-1}(\mathbf{t})$ is stable by the action of $T_N$. If $\mathbf{t}$ is a regular value of $\mu$, then $\mathrm{Lie}(T)$ acts infinitesimally freely on $\mu^{-1}(\mathbf{t})$. Thus $X_\mathbf{t} = \mu^{-1}(\mathbf{t})/T$ is an orbifold with a $T_N/T$ action, provided with a symplectic form $\omega_\mathbf{t}$. Its (real) dimension $d$ is the even integer $\dim X - 2r$. The equivariant symplectic form of $X_\mathbf{t}$ is $\omega_\mathbf{t}(\epsilon) = \omega_\mathbf{t} + \langle \tilde{\mu}(m), \epsilon \rangle$. One defines

$$\mathcal{F}_\epsilon(\mathbf{t}) = \frac{1}{(2i\pi)^{d/2}} \int_{X_\mathbf{t}} e^{i\omega_\mathbf{t}(\epsilon)} = \frac{1}{(2i\pi)^{d/2}} \int_{X_\mathbf{t}} e^{i\omega_\mathbf{t}} e^{i\langle \tilde{\mu}(m), \epsilon \rangle} \, .$$

If $\tilde{\mu}(m)$ growths sufficiently fast at $\infty$, then $\mathcal{F}_\epsilon(\mathbf{t})$ is well defined as a generalized function of $\epsilon$. This is verified in the case $X = \mathbb{C}^N$ considered by NPZ. When $X_\mathbf{t}$ is compact, the value at $\epsilon = 0$ of $\mathcal{F}_\epsilon(\mathbf{t})$ is the volume of $X_\mathbf{t}$ for the symplectic form $\omega_\mathbf{t}/(2\pi)$. So the generalized function $\epsilon \mapsto \mathcal{F}_\epsilon(\mathbf{t})$ is called the equivariant volume of $X_\mathbf{t}$.

Let $\mathfrak{c} \subseteq \mathrm{Lie}(T)^*$ be an open connected subset contained in the set of regular values of $\mu$, and let $X_\mathfrak{c} = \mu^{-1}(\mathfrak{c})$. If $m \in \mathrm{Lie}(T_N)^*$, it defines a linear function $\langle m, \epsilon \rangle$ on $\mathrm{Lie}(T_N)$, thus can be considered as a $T_N$ closed equivariant form on $X_\mathfrak{c}$ of equivariant degree 2. We say that $m$ is $\mu$-compact if the class of $\langle m, \epsilon \rangle$ in equivariant cohomology is equal to the class of a closed equivariant form $\mathrm{Th}(m)(\epsilon)$ on $X_\mathfrak{c}$ such that $\mathrm{Th}(m)(\epsilon)$ restricted to $\mu^{-1}(\mathbf{t})$ ($\mathbf{t} \in \mathfrak{c}$) is compactly supported. We give a simple example in the last section.

Here is the shift equation of NPZ.

**Proposition C.1.** *Let $m \in \mathrm{Lie}(T_N)^*$ be $\mu$-compact. Let $\mathbf{t} \in \mathfrak{c}$ be a regular value of $\mu$ and assume that $m$ is sufficiently small so that $\mathbf{t} + sm$ is a regular value of $\mu$ for $s \in [0,1]$. Then*

$$\mathcal{F}_\epsilon(\mathbf{t}) - e^{-i\langle m, \epsilon \rangle} \mathcal{F}_\epsilon(\mathbf{t} + \Psi(m))$$

*is an analytic function of $\epsilon$.*

In the following, we prove this statement by giving a compactly supported integral formula for

$$N_\epsilon(\mathbf{t}, m) = \mathcal{F}_\epsilon(\mathbf{t}) - e^{-i\langle m, \epsilon \rangle} \mathcal{F}_\epsilon(\mathbf{t} + \Psi(m)) \, .$$

Consider $X_\mathfrak{c} = \mu^{-1}(\mathfrak{c})$. It is provided with a locally free action of $T$. A basic (with respect to $T$) equivariant form on $X_\mathfrak{c}$ is a $T_N$-equivariant form $\nu(\epsilon)$ depending only of $\epsilon$ modulo $\mathrm{Lie}(T)$ ($\nu(\epsilon + u) = \nu(\epsilon)$ if $u \in \mathrm{Lie}(T)$), and such that $\iota(u_X)\nu(\epsilon) = 0$ for any $u \in \mathrm{Lie}(T)$. If $T$ acts freely, basic forms are pull back of $T_N/T$-equivariant forms on $X_\mathfrak{c}/T$. We still say that basic forms are $T_N/T$-equivariant forms on $X_\mathfrak{c}/T$ when the action is only infinitesimally free. The ring of $T$-basic equivariant forms is stable by the equivariant differential $D_\epsilon$ and the pull back map induces an isomorphism $H^*_{T_N/T}(X_\mathfrak{c}/T) \to H^*_{T_N}(X_\mathfrak{c})$ of the equivariant cohomology rings (this is essentially due to H. Cartan [42]). For integration on non compact spaces, we need a description in terms of equivariant forms. Choose a $T_N$ invariant connection form $\alpha \in \Omega^1(X_\mathfrak{c}) \otimes \mathrm{Lie}(T)$ on $X_\mathfrak{c}$ for the infinitesimally free action of $\mathrm{Lie}(T)$. So $\alpha(\epsilon_X) = \epsilon$ if $\epsilon \in \mathrm{Lie}(T)$. Then there is an homomorphism $W_\alpha$ from the ring of $T_N$- equivariant forms on $X_\mathfrak{c}$ to $T_N/T$-equivariant forms on $X_\mathfrak{c}/T$ commuting with the equivariant differential. The map $W_\alpha$ is a generalization of the Chern-Weil homomorphism (see Lemma 23 in [43], or Proposition 85 in [44]). It is described as follows. Let $R_\alpha = d\alpha$ be the curvature of $\alpha$, and $R_\alpha(\epsilon) = d\alpha - \langle \alpha, \epsilon_X \rangle$ be its equivariant curvature (an element of $\mathrm{Lie}(T)$ with coefficient differential forms on $X_\mathfrak{c}$). The homomorphism $W_\alpha$ consists in replacing in an equivariant differential form the variable $\epsilon \in \mathrm{Lie}(T_N)$ by $\epsilon + R_\alpha(\epsilon)$, and project the resulting differential form on horizontal forms.

In particular,

$$W_\alpha(m)(\epsilon) = \langle m, \epsilon \rangle + \langle \Psi(m), R_\alpha(\epsilon) \rangle$$

is closed and basic. Indeed $W_\alpha(m)(\epsilon + u) = W_\alpha(m)(\epsilon)$ when $u \in \mathrm{Lie}(T)$ since $R_\alpha(u) = R_\alpha - u$ for $u \in \mathrm{Lie}(T)$. For $\epsilon = 0$, $W_\alpha(m)(0) = \langle \Psi(m), R_\alpha \rangle$. If $m$ is $\mu$-compact and equivalent to $\mathrm{Th}(m)(\epsilon)$, define $\mathrm{Th}_\alpha(m)(\epsilon)$ to be the image by $W_\alpha$ of $\mathrm{Th}(m)(\epsilon)$. Then $\mathrm{Th}_\alpha(m)(\epsilon)$ restricted to $X_{\mathbf{t}}$ is a closed equivariant form which is compactly supported and $\mathrm{Th}_\alpha(m)(0)$ is a compactly supported closed two-form on $X_{\mathbf{t}}$.

Let $\mathbf{t}_1 \in \mathfrak{c}$ and let $X_1 = \mu^{-1}(\mathbf{t}_1)/T$, with symplectic form $\omega_1$. We denote by $\omega_1(\epsilon)$ the equivariant symplectic form of $X_1$. So $\omega_1 = \omega_1(0)$.

**Proposition C.2.** *If $\mathbf{t}$ is near $\mathbf{t}_1$ and $m$ is small, then we have the compactly supported integral representation of $N_\epsilon(\mathbf{t}, m)$:*

$$N_\epsilon(\mathbf{t}, m) = \frac{i}{(2i\pi)^{d/2}} \int_{X_1} \int_{s=0}^1 \mathrm{Th}_\alpha(m)(\epsilon) e^{i(\omega_1(\epsilon) - \langle \mathbf{t} - \mathbf{t}_1, R_\alpha(\epsilon) \rangle)} e^{-is W_\alpha(m)(\epsilon)} ds \,.$$

*The value of $N_\epsilon(\mathbf{t}, m)$ at $\epsilon = 0$ is*

$$\frac{1}{(2\pi)^{d/2}} \sum_{j>0, k\geq 0; j+k=d/2} \int_{X_1} (-1)^j \frac{(\omega_1 - \langle \mathbf{t} - \mathbf{t}_1, R_\alpha \rangle)^k}{k!} \frac{\mathrm{Th}_\alpha(m)(0)^j}{j!} \,.$$

*Proof.* Let us sketch a proof of the above proposition. We give an example at the end in the simplest case $A_1$, in order to see how arguments on convergence may be justified.

We work in a neighborhood of the regular value $\mathbf{t}_1$ of $\mu$, so that all the spaces $X_{\mathbf{t}}$ are isomorphic to the same manifold $X_1$. Let $P = \mu^{-1}(\mathbf{t}_1)$. So we may assume that $X = P \times \mathrm{Lie}(T)^*$, where $P$ is provided with a locally free action of $T$ (and still acted by the larger torus $T_N$), the moment map $\mu$ being the second projection.

Let $p_1 : X \to P$ be the first projection. We choose a $T_N$-invariant connection form $\alpha$ for the action of $T$ on $P$ and still denote by $\alpha$ the reciproc image of $\alpha$ on $X$. Let us analyze the $T_N$-equivariant symplectic form $\omega(\epsilon)$ in this system of coordinates $P \times \mathrm{Lie}(T)^*$.

**Lemma C.3.** *There exists a $T$-basic form $\nu \in \Omega^1(X)$, invariant by $T_N$, such that for $\epsilon \in \mathrm{Lie}(T_N)$:*

$$\omega(\epsilon) = p_1^* \omega_1(\epsilon) - \langle \mathbf{t} - \mathbf{t}_1, R_\alpha(\epsilon) \rangle - \langle d\mathbf{t}, \alpha \rangle + D_\epsilon \nu \,.$$

*Proof.* The form $\omega_\alpha(\epsilon) = \langle \mathbf{t} - \mathbf{t}_1, R_\alpha(\epsilon) \rangle + \langle d\mathbf{t}, \alpha \rangle$ is equivarianty closed since this is $D_\epsilon \langle \mathbf{t} - \mathbf{t}_1, \alpha \rangle$. Similarly $p_1^* \omega_1(\epsilon)$ is equivariantly closed. Now $\omega(\epsilon) - (p_1^* \omega_1(\epsilon) - \omega_\alpha(\epsilon))$ restricts to $P$ by zero. Using the standard homotopy for the contraction $p_1 : P \times \mathrm{Lie}(T)^* \to P$, we see that there exists a $T_N$ invariant 1- form $\nu$ on $P \times \mathrm{Lie}(T)^*$ such that

$$\omega(\epsilon) = p_1^* \omega_1(\epsilon) - (\langle \mathbf{t} - \mathbf{t}_1, R_\alpha(\epsilon) \rangle + \langle d\mathbf{t}, \alpha \rangle) + D_\epsilon \nu \,.$$

Let us see that $\nu$ is basic for the action of $T$, that is $\nu(\epsilon_X) = 0$ if $\epsilon \in \mathrm{Lie}(T)$. Compute the function term (that is the degree 0 term with respect to the degree of differential forms) in the equation above when $\epsilon \in \mathrm{Lie}(T)$. The degree 0 term of $D_\epsilon \nu$ is $-\langle \nu, \epsilon_X \rangle$. The degree 0 term of $\omega(\epsilon) - p_1^* \omega_1(\epsilon)$ is $\langle \mathbf{t} - \mathbf{t}_1, \epsilon \rangle$ and the degree 0 term of $\langle \mathbf{t} - \mathbf{t}_1, R_\alpha(\epsilon) \rangle$ is $-\langle \mathbf{t} - \mathbf{t}_1, \epsilon \rangle$. So $\langle \nu, \epsilon_X \rangle = 0$. $\square$

Identify $X_{\mathbf{t}}$ with $X_1$ by the first projection. From the lemma above, we obtain the following corollary.

**Corollary C.4.** *On $X_1$, we have the equation*

$$\omega_{\mathbf{t}}(\epsilon) = \omega_1(\epsilon) - \langle \mathbf{t} - \mathbf{t}_1, R_\alpha(\epsilon) \rangle + D_\epsilon \nu_{\mathbf{t}} \,,$$

*where $\nu_{\mathbf{t}}$ is the one form on $X_1 = P/T$ deduced from the basic one form $\nu | P \times \{\mathbf{t}\}$ by quotient.*

So we see that the cohomology class of $\omega_{\mathbf{t}}(\epsilon)$ varies linearly with $\mathbf{t}$. This is the content (when $T = T_N$) of Duistermaat-Heckman theorem [45].

Using the fact that integrals of $D_\epsilon$-exact equivariant forms integrated against test functions vanishes, we conclude that

$$\mathcal{F}_\epsilon(\mathbf{t}) = \frac{1}{(2i\pi)^{d/2}} \int_{X_1} e^{i\omega_{\mathbf{t}}(\epsilon)} = \frac{1}{(2i\pi)^{d/2}} \int_{X_1} e^{i\omega_1(\epsilon)} e^{-i\langle \mathbf{t}-\mathbf{t}_1, R_\alpha(\epsilon)\rangle}.$$

Define

$$\mathcal{F}(s, \epsilon, \mathbf{t}) = e^{-is\langle m, \epsilon\rangle} \mathcal{F}_\epsilon(\mathbf{t} + s\Psi(m))$$

for $s \in [0, 1]$. So $\mathcal{F}(0, \epsilon, \mathbf{t}) - \mathcal{F}(1, \epsilon, \mathbf{t}) = N_\epsilon(\mathbf{t}, m)$ is the shift we want to compute.

We relate it to the generalized Chern Weil homomorphism. Indeed

$$\mathcal{F}(s, \epsilon, \mathbf{t}) = \frac{1}{(2i\pi)^{d/2}} \int_{X_1} e^{i(\omega_1(\epsilon) - \langle \mathbf{t}-\mathbf{t}_1, R_\alpha(\epsilon)\rangle)} e^{-is\langle m, \epsilon\rangle} e^{-is\langle \Psi(m), R_\alpha(\epsilon)\rangle}$$

$$= \frac{1}{(2i\pi)^{d/2}} \int_{X_1} e^{i(\omega_1(\epsilon) - \langle \mathbf{t}-\mathbf{t}_1, R_\alpha(\epsilon)\rangle)} e^{-isW_\alpha(m)(\epsilon)},$$

since $W_\alpha(m)(\epsilon) = \langle m, \epsilon\rangle + \langle \Psi(m), R_\alpha(\epsilon)\rangle$. So we obtain that $\frac{d}{ds}\mathcal{F}(s, \epsilon, \mathbf{t})$ is equal to

$$\frac{-i}{(2i\pi)^{d/2}} \int_{X_1} W_\alpha(m)(\epsilon) e^{i(\omega_1(\epsilon) - \langle \mathbf{t}-\mathbf{t}_1, R_\alpha(\epsilon)\rangle)} e^{-isW_\alpha(m)(\epsilon)}. \tag{C.1}$$

This equation holds true without any hypothesis on $m$.

Now assume that $m$ is $\mu$-compact. So we can replace $W_\alpha(m)(\epsilon)$ by $\mathrm{Th}_\alpha(m)(\epsilon)$. Integrating in $s$, we obtain the first assertion of Proposition C.2.

Let us compute the value at $\epsilon = 0$. Then $W_\alpha(m)(0) = \langle \Psi(m), R_\alpha\rangle$ is a differential form of degree 2. So

$$\int_{s=0}^{1} e^{-isW_\alpha(m)(0)} = \frac{e^{-iW_\alpha(m)(0)} - 1}{-iW_\alpha(m)(0)} = \sum_{j=0}^{d/2} (-i)^j \frac{W_\alpha(m)(0)^j}{(j+1)!}.$$

We can replace $W_\alpha(m)(0)$ by $\mathrm{Th}_\alpha(m)(0)$ which is equivalent in cohomology. We then obtain the formula for the value at $\epsilon = 0$. $\qquad\square$

Assume that $m_1, m_2, \ldots, m_J$ is a sequence of $\mu$-compact elements, then $m = \sum_{k=1}^{J} s_k m_k$ is $\mu$-compact. If $\mathbf{t}$ is sufficiently near $\mathbf{t}_1$ and $s_k$ are sufficiently small, then

$$N_\epsilon(\mathbf{t}, \sum_k s_k m_k) = \mathcal{F}_\epsilon(\mathbf{t}) - e^{-i\sum_k s_k\langle m_k, \epsilon\rangle} \mathcal{F}_\epsilon(\mathbf{t} + \sum_k s_k\Psi(m_k))$$

is an analytic function of $\epsilon$. We see that from the formula above for $m = \sum_k s_k m_k$ that the value $N_\epsilon(\mathbf{t}, \sum s_k m_k))$ at $\epsilon = 0$ is a polynomial in $\mathbf{t}$ and $s_k$ of total degree $d/2$. It involves products of the forms $\mathrm{Th}_\alpha(m_k)$. In the case considered by the authors, the $m_k, k = 1, \ldots, J$ corresponds to compact divisors $D_k$ of the non compact toric manifold $X_1$, and therefore their product are obtained by computing various cohomological intersections of the compact divisors $D_k$.

Consider a product of linear forms $\prod_{i\in I} m_i$. It can happen that the closed equivariant form $\prod_{i\in I}\langle m_i, \epsilon\rangle$ is cohomologous on $X_{\mathfrak{c}}$ to a form $\mathrm{Th}_I(\epsilon)$ such that $\mathrm{Th}_I(\epsilon)$ restricted to $\mu^{-1}(\mathbf{t})$ is compactly supported, but each individual $m_i$ itself non compact (see Example 6.1 of the NPZ article). Using the formula

$$\prod_{i\in I}(e^{x_i} - 1) = \sum_{J\subset I} (-1)^{|J|} e^{\sum_{j\in J} x_j} - 1 = (\prod_{i\in I} x_i) \prod_{i\in I} \frac{e^{x_i} - 1}{x_i},$$

one obtains the generalized shift equation of NPZ when $\prod_{i\in I}\langle m_i, \epsilon\rangle$ is $\mu$-compact:

$$N_\epsilon(\mathbf{t}, I) = \sum_{J\subset I}(-1)^{|J|}e^{i\sum_{j\in J}\langle m_j, \epsilon\rangle}\mathcal{F}_\epsilon(\mathbf{t}+\sum_{j\in J}\Psi(m_j))$$

is analytic in $\epsilon$.

## C.2 A simple example

We reconsider the example 5.1 ($A_1$-space) of the NPZ article.

Let $X = \mathbb{C}^3$ with standard action of $T_3 = U(1)^3$. We write $\epsilon\in\mathrm{Lie}(T_3)$ as $\epsilon_1 p^1 + \epsilon_2 p^2 + \epsilon_3 p^3$, $p_X^i$ being the vector field $y_i\partial_{x_i} - x_i\partial_{y_i}$. The equivariant symplectic form is

$$\omega(\epsilon) = (dx_1\wedge dy_1 + dx_2\wedge dy_2 + dx_3\wedge dy_3) + \frac{1}{2}(\epsilon_1|z_1|^2 + \epsilon_2|z_2|^2 + \epsilon_3|z_3|^2).$$

If we integrate (in $\epsilon$) the function $e^{i(\epsilon_1|z_1|^2 + \epsilon_2|z_2|^2 + \epsilon_3|z_3|^2)/2}$ against a smooth compactly supported function of $\epsilon$, the result is a rapidly decreasing function in $z_1, z_2, z_3$. So $\frac{1}{(2i\pi)^3}\int_{\mathbb{C}^3}e^{i\omega(\epsilon)}$ is well defined as a generalized function of $\epsilon\in\mathrm{Lie}(T_3)$. It is the Fourier transform of the characteristic function of the cone in $\mathrm{Lie}(T_3)^*$ generated by $p_1, p_2, p_3$. Outside the hyperplanes $\epsilon_i = 0$, it is given by the rational function $\frac{1}{\epsilon_1\epsilon_2\epsilon_3}$.

Let $T = U(1)$ and let $T\to T_3$ be the homomorphism associated to

$$Q = \begin{pmatrix} 1 & -2 & 1 \end{pmatrix}.$$

We denote by $J$ the generator of $\mathrm{Lie}(T)$, with $J_X = (y_1\partial_{x_1} - x_1\partial_{y_1}) - 2(y_2\partial_{x_2} - x_2\partial_{y_2}) + (y_3\partial_{x_3} - x_3\partial_{y_3})$. The moment map $\mu : X\to\mathrm{Lie}(T)^*$ for the action of $T$ is $\mu(z_1, z_2, z_3) = \frac{1}{2}(|z_1|^2 - 2|z_2|^2 + |z_3|^2)J^*$ with $\langle J^*, J\rangle = 1$.

Let $t > 0$ and let

$$P_t = \{(z_1, z_2, z_3)\in\mathbb{C}^3; \frac{1}{2}(|z_1|^2 - 2|z_2|^2 + |z_3|^2) = t\},$$

which is not compact, since $z_2$ can be arbitrary large. Then $X_t = P_t/T$ is a smooth non compact manifold of real dimension 4 (the total space of $\mathcal{O}(-2)\to\mathbb{P}_1(\mathbb{C})$).

Let $\omega_t(\epsilon)$ be the equivariant symplectic form of $X_t$ obtained from $\omega(\epsilon)$ by restriction and quotient. Then

$$\mathcal{F}_\epsilon(tJ^*) = \frac{1}{(2i\pi)^2}\int_{X_t}e^{i\omega_t(\epsilon)}$$

is a generalized function of $\epsilon$. It is the Fourier transform of the characteristic function of the (non compact) polyhedron $C_t = \{p_1 - 2p_2 + p_3 = t, p_1\geq 0, p_2\geq 0, p_3\geq 0\}$ with $p_i = \frac{|z_i|^2}{2}$. Outside the hyperplanes $\epsilon_1 - \epsilon_3 = 0, 2\epsilon_1 + \epsilon_2 = 0, 2\epsilon_3 + \epsilon_2 = 0$ it is given by

$$\frac{e^{it\epsilon_1}}{(2\epsilon_1 + \epsilon_2)(\epsilon_1 - \epsilon_3)} + \frac{e^{it\epsilon_3}}{(2\epsilon_3 + \epsilon_2)(\epsilon_3 - \epsilon_1)}.$$

We will see that the linear form $p_2(\epsilon) = \epsilon_2$ is $\mu$-compact, reflecting the fact that $C_t\cap\{p_2 = 0\} = \{p_2 = 0, p_1 + p_3 = t, p_1\geq 0, p_3\geq 0\}$ is compact. We have $\Psi(-mp_2) = 2mJ^*$. The shift equation of NPZ (for $-p_2$) says that, when $t > 0$ and $t + 2m > 0$,

$$N_\epsilon(t) = \mathcal{F}_\epsilon(tJ^*) - e^{im\epsilon_2}\mathcal{F}_\epsilon((t + 2m)J^*)$$

is an analytic function of $\epsilon_1, \epsilon_2, \epsilon_3$ with value at 0 equal to $m(t + m)$. Of course in this example, everything can be computed directly. However we sketch below a complicated proof, but it clarifies the local equivariant cohomology arguments given in the proof of Proposition C.1.

**First**. We identify all the manifolds $X_t$, for $t > 0$, to the same manifold $X_1 = P_1/T$ using homothety, and denote still by $\omega_t(\epsilon)$ the corresponding equivariant symplectic form on $X_1$. Then $\omega_t(\epsilon) = t\omega_1(\epsilon)$ is already in the form provided by Duistermaat-Heckman theorem (linear dependance in $t$). So $\mathcal{F}_\epsilon(tJ^*)$ is equal to

$$\frac{1}{(2i\pi)^2} \int_{X_1} e^{it\omega_1(\epsilon)}.$$

Define

$$\mathcal{F}(s,\epsilon,t) = \frac{1}{(2i\pi)^2} \int_{X_1} e^{ism\epsilon_2} e^{i(t+2ms)\omega_1(\epsilon)}.$$

Here $s$ vary between 0 and 1. Thus $N_\epsilon(t) = \mathcal{F}(0,\epsilon,t) - \mathcal{F}(1,\epsilon,t)$.

**Second**. Our next step is to compute $\frac{d}{ds}\mathcal{F}(s,\epsilon,t)$ using the generalized Chern Weil homomorphism $W_\alpha$ associated to a connection form on $X_1$.

Let

$$\alpha = -\frac{1}{2}(x_1 dy_1 - y_1 dx_1 + x_2 dy_2 - y_2 dx_2 + x_3 dy_3 - y_3 dx_3)J,$$

which restricts to $P_1$ as a connection form with value in $\text{Lie}(T) = \mathbb{R}J$. Let $R_\alpha$ be its curvature, and $R_\alpha(\epsilon) = d\alpha - \langle \alpha, \epsilon_X \rangle$ be its equivariant curvature with value in $\text{Lie}(T)$. The equivariant curvature $R_\alpha(\epsilon)$ of $\alpha$ restricted to $P_1$ is $-\omega_1(\epsilon)J$ with

$$\omega_1(\epsilon) = \omega_1 + \frac{1}{2}(\epsilon_1|z_1|^2 + \epsilon_2|z_2|^2 + \epsilon_3|z_3|^2).$$

The Chern Weil homomorphism $W_\alpha$ consists in replacing in an equivariant form the variable $\epsilon = \sum \epsilon_i p^i$ by

$$\epsilon + R_\alpha(\epsilon) = (\epsilon_1 - \omega_1(\epsilon))p^1 + (\epsilon_2 + 2\omega_1(\epsilon))p^2 + (\epsilon_3 - \omega_1(\epsilon))p^3$$

and project the result on basic forms. For example, consider $\langle p_2, \epsilon \rangle = \epsilon_2$ as a closed equivariant form on $X = \mathbb{C}^3$. Then

$$W_\alpha(p_2)(\epsilon) = \epsilon_2 + 2\omega_1(\epsilon),$$

a closed equivariant form on $\mathbb{C}^3$. We have $W_\alpha(p_2)(\epsilon) = (\epsilon_1 + \epsilon_2/2)|z_1|^2 + (\epsilon_3 + \epsilon_2/2)|z_3|^2 + 2\omega_1$ on $P_1$. It depends only of $\epsilon$ modulo $\text{Lie}(T))$, and it is already horizontal. So it defines an $T_N/T$-equivariant closed form on $X_1$. Thus, we can rewrite

$$\mathcal{F}(s,\epsilon,t) = \frac{1}{(2i\pi)^2} \int_{X_1} e^{ism\epsilon_2} e^{i(t+2ms)\omega_1(\epsilon)} = \frac{1}{(2i\pi)^2} \int_{X_1} e^{it\omega_1(\epsilon)} e^{imsW_\alpha(p_2)(\epsilon)}.$$

Equation C.1 says that as a generalized function of $\epsilon$, we have

$$\frac{d}{ds}\mathcal{F}(s,\epsilon,t) = \frac{im}{(2i\pi)^2} \int_{X_1} W_\alpha(p_2)(\epsilon) e^{it\omega_1(\epsilon)} e^{imsW_\alpha(p_2)(\epsilon)},$$

which is readily verified.

**Third**. We now prove and use the fact that $p_2$ is $\mu$-compact. This will allow us to give a convergent integral expression for

$$N_\epsilon(t) = -\int_{s=0}^1 \frac{d}{ds}\mathcal{F}(s,\epsilon,t).$$

Let $A(x)$ be a smooth function on $\mathbb{R}$, supported in a small neighborhood if 0, and equal to 1 at 0. Let $A'$ its derivative. We consider

$$P_2(\epsilon) := \epsilon_2 A(\frac{x_2^2 + y_2^2}{2}) + A'(\frac{x_2^2 + y_2^2}{2})dx_2 \wedge dy_2.$$

It is equivariantly closed on $\mathbb{C}^3$ for $D_\epsilon$ and equivalent to $p_2(\epsilon)$ in cohomology. Indeed one verifies that

$$P_2(\epsilon) - \epsilon_2 = D_\epsilon v,$$

with

$$v = \frac{1}{2}\left(\frac{A(\frac{x_2^2+y_2^2}{2}) - 1}{\frac{x_2^2+y_2^2}{2}}\right)(x_2 dy_2 - y_2 dx_2).$$

Remark that

$$\frac{1}{2\pi}\int_{x_2,y_2} P_2(\epsilon) = \frac{1}{2\pi}\int_0^\infty A'(r)dr d\theta = -1,$$

since $A(0) = 1$. In other words $-P_2(\epsilon)$ is the reciproc image of the equivariant Thom form of $\mathbb{C}$ by the fibration $U : \mathbb{C}^3 \to \mathbb{C}$ given by $(z_1, z_2, z_3) \to z_2$.

The support of $P_2(\epsilon)$ intersects the fiber of $\mu$ in a compact set, so $p_2$ is $\mu$-compact. So $W_\alpha(P_2)(\epsilon)$ will be a compactly supported form on $X_1$ equivalent to $W_\alpha(p_2)(\epsilon)$. Let us compute $W_\alpha(P_2)(\epsilon) = \mathrm{Th}_2(\epsilon)$. By definition, $\mathrm{Th}_2(\epsilon)$ is the horizontal projection of $(\epsilon_2 + 2\omega_1(\epsilon))A(\frac{x_2^2+y_2^2}{2}) + A'(\frac{x_2^2+y_2^2}{2})dx_2 \wedge dy_2$ on $P$. The horizontal projection of $dx_2 \wedge dy_2$ is

$$dx_2 \wedge dy_2 + (\iota(J_X)(dx_2 \wedge dy_2)) \wedge \alpha = dx_2 \wedge dy_2 - 2(x_2 dx_2 + y_2 dy_2) \wedge \alpha.$$

Consider the two form

$$\Gamma_2 = 2A(\frac{x_2^2+y_2^2}{2})\omega_1 + A'(\frac{x_2^2+y_2^2}{2})dx_2 \wedge dy_2 - 2A'(\frac{x_2^2+y_2^2}{2})(x_2 dx_2 + y_2 dy_2) \wedge \alpha.$$

Thus $d\Gamma_2 = 0$ and

$$\mathrm{Th}_2(\epsilon) = ((\epsilon_1 + \epsilon_2/2)|z_1|^2 + (\epsilon_3 + \epsilon_2/2)|z_3|^2)A(\frac{x_2^2+y_2^2}{2}) + \Gamma_2$$

on $P$. We have

$$W_\alpha(v) = v - \frac{1}{2}\left(A(\frac{x_2^2+y_2^2}{2}) - 1\right)\alpha.$$

It is easy to verify that indeed $\mathrm{Th}_2(\epsilon)$ is equivariantly closed, basic and that $\mathrm{Th}_2(\epsilon) - W_\alpha(p_2) = D_\epsilon W_\alpha(v)$. The boundary term $D_\epsilon W_\alpha(v)$ does not contribute to the integral, since integrated against a test function, it gives a differential form on $X_1$ rapidly decreasing at $\infty$, as easy to check. We have the equation

$$\frac{d}{ds}\mathcal{F}(s,\epsilon,t) = \frac{im}{(2i\pi)^2}\int_{X_1} \mathrm{Th}_2(\epsilon)e^{ism\epsilon_2}e^{i(t+2ms)\omega_1(\epsilon)}.$$

This shows that

$$N_\epsilon(t) = \frac{-im}{(2i\pi)^2}\int_{P/T}\int_{s=0}^1 \mathrm{Th}_2(\epsilon)e^{it\omega_1(\epsilon)}e^{isW_\alpha(p_2)(\epsilon)},$$

so $N_\epsilon(t) = \mathcal{F}_\epsilon(tJ^*) - e^{im\epsilon_2}F_\epsilon((t+2m)J^*)$ is analytic in $\epsilon$.

Consider $\epsilon = 0$ in this formula. So $N_{\epsilon=0}(t)$ is equal to

$$-m \int_{s=0}^{1} (t + 2ms)ds \left( \frac{1}{(2\pi)^2} \int_{P/T} \omega_1 \wedge \text{Th}_2(0) \right) = m(t + m),$$

provided we prove that $\int_{P/T} \omega_1 \wedge (-\text{Th}_2(0)) = (2\pi)^2$. As we have seen, the form $\frac{-1}{2\pi} P_2(\epsilon)$ is the equivariant Thom form of the normal bundle of $\mathbb{C}^3 \cap z_2 = 0$ in $\mathbb{C}^3$. So $\frac{-1}{2\pi} W_\alpha(P_2)(\epsilon)$ is the equivariant Thom form of the normal bundle of $(P \cap \{z_2 = 0\})/T$ in $P/T$. So we obtain that $\frac{-1}{(2\pi)^2} \int_{P/T} \omega_1 \wedge \text{Th}_2(0) = \frac{1}{2\pi} \int_{(P \cap \{z_2=0\})/T} \omega_1$. But $(P \cap \{z_2 = 0\})/T$ is $\{|z_1|^2/2 + |z_3|^2/2 = 1\}/T$, and has volume $2\pi$ for $\omega_1$. So we obtain our constant 1.

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
