# Peer review of "Shifts of prepotentials (with an appendix by Michele Vergne)"

_SciPost Physics, doi:SciPost Phys. 12, 177 (2022)_

## Round 1 · Referee Report · Anonymous (Referee 1) · 2021-12-23

Strengths

1- A timely solution to the problem of rigorously computing the triple intersection numbers in non-compact (toric) Calabi-Yau manifolds using localisation methods. 2- Well-written and well-motivated introduction to the subject and to the main results of the paper. 3- Clear presentation of the set up and the methods, and well explained examples to illustrate the main results.

Weaknesses

  • It lacks a clear conclusion that summarises the results and offers general perspectives for future works.

Report

M-theory compactification on a Calabi-Yau $3$-fold $X$ in the decompactification gives rise to a five dimensional $N=1$ supersymmetric field theory. The prepotential of the supersymmetric field theory is given by triple intersection numbers of divisors in $X$ and it can be computed formally by the integration of the exponential of the Kahler form over $X$. However in the decompactification limit, the integration over $X$ is ill-defined, and there has been lacking for a long time a rigorous way to define and to compute the triple intersection numbers in a non-compact Calabi-Yau.

The authors of this paper solve this problem by arguing that when the non-compact Calabi-Yau $X$ is toric and enjoys many isometries, one can use the Duistermaat-Heckman localisation formula to define an equivariant volume over $X$ that regularises the non-compactness of the Calabi-Yau. Even though the equivariant volume is singular in the limit the equivariant parameters are sent to zero, it satisfies a difference equation, which is not singular in this limit, and from which one can read off the regularised triple intersection numbers.

The authors also upgrade the results by including quantum corrections, which allows extraction of genus one constant map contribution to the Calabi-Yau in topological string as well. The authors discuss their methods (difference equations) for both the cases where $H^2_{\text{comp}}(X)$ is non-trivial and $H^2_{\text{comp}}(X)=0$ but $H^4_{\text{comp}}(X)$ is non-trivial, and give many examples to illustrate their methods. They also generalise their results to Calabi-Yau $5$-folds by considering a fibered product of $X$ and $\mathbb{C}^2$ and use this result to prove an important identity in one of their previous papers that is concerned with higher rank Donaldson-Thomas theory on Calabi-Yau threefolds.

This paper solves a long-standing problem in mathematical physics in a rigorous way. It is clearly written, with a well-motivated introduction to the subject and to the main results of the paper. It presents clearly the set up, and the methods, and it illustrates the methods with numerous examples. Although it lacks a clear conclusion that summarises the results and offers general perspectives for future works. In general, I recommend its publication in this journal after a conclusion is included.

Requested changes

  • Give a clear conclusion that summarises the results and offers general perspectives for future works.

---

## Round 2 · Referee Report · Anonymous · 2022-2-8

Report

The new version of the draft fulfils the requested changes by including a succinct conclusion that clearly summarises the results of this paper. Better still, a new appendix is added to give an alternative proof to the shift equation, further strengthening the results of this paper. The publication of this paper is therefore highly recommended.

---

## Round 2 · Referee Report · Anonymous · 2022-4-14

Strengths

This paper addresses an interesting and basic problem in topological string theory which was not clear before

Weaknesses

No weaknesses

Report

The classical limit of the prepotential in topological string theory involves the triple intersection numbers of the Calabi-Yau threefold. However, when this threefold is non-compact, as it happens in local mirror symmetry, these intersection numbers are not really well defined. This paper studies this problem from the point of view of \epsilon-regularized volumes and the DH formula and proposes a mathematical framework to address this issue.

---

## Round 2 · Author Response

Dear Editor and Referee,

thanks for your suggestions, which we implemented in v2.
Moreover, an appendix by M. Vergne has been added,
providing an alternative local proof of the shift equation.

Best regards,
Nicolo Piazzalunga

---

## Round 2 · List of Changes

- add conclusion section
- add appendix by M. Vergne
- fix a few typos and clarify notation

---

## Editorial Decision

published